# IMPLICIT REGULARIZATION OF DEEP RESIDUAL NETWORKS TOWARDS NEURAL ODES

**Pierre Marion**,* **Yu-Han Wu**\*
LPSM
Sorbonne Université, CNRS
Paris, France

**Michael E. Sander**
DMA
ENS, CNRS
Paris, France

**Gérard Biau**
LPSM
Sorbonne Université, CNRS
Paris, France

## ABSTRACT

Residual neural networks are state-of-the-art deep learning models. Their continuous-depth analog, neural ordinary differential equations (ODEs), are also widely used. Despite their success, the link between the discrete and continuous models still lacks a solid mathematical foundation. In this article, we take a step in this direction by establishing an implicit regularization of deep residual networks towards neural ODEs, for nonlinear networks trained with gradient flow. We prove that if the network is initialized as a discretization of a neural ODE, then such a discretization holds throughout training. Our results are valid for a finite training time, and also as the training time tends to infinity provided that the network satisfies a Polyak-Łojasiewicz condition. Importantly, this condition holds for a family of residual networks where the residuals are two-layer perceptrons with an overparameterization in width that is only linear, and implies the convergence of gradient flow to a global minimum. Numerical experiments illustrate our results.

## 1  INTRODUCTION

Residual networks are a successful family of deep learning models popularized by breakthrough results in computer vision (He et al., 2016b). The key idea behind residual networks, namely the presence of skip connections, is now ubiquitous in deep learning, and can be found, for example, in Transformer models (Vaswani et al., 2017). The main advantage of skip connections is to allow successful training with depth of the order of a thousand layers, in contrast to vanilla neural networks, leading to significant performance improvement (e.g., Wang et al., 2022). This has motivated research on the properties of residual networks in the limit where the depth tends to infinity. One of the main explored directions is the neural ordinary differential equation (ODE) limit (Chen et al., 2018).

Before presenting neural ODEs, we first introduce the mathematical formalism of deep residual networks. We consider a single model throughout the paper to simplify the exposition, but most of our results apply to more general models, as will be discussed later. We consider the formulation

$$h_{k+1} = h_k + \frac{1}{L\sqrt{m}} V_{k+1} \sigma\Big(\frac{1}{\sqrt{q}} W_{k+1} h_k\Big), \quad k \in \{0, \dots, L-1\}, \tag{1}$$

where $L$ is the depth of the network, $h_k \in \mathbb{R}^q$ is the output of the $k$-th hidden layer, $V_k \in \mathbb{R}^{q \times m}$, $W_k \in \mathbb{R}^{m \times q}$ are the weights of the $k$-th layer, and $\sigma$ is an activation function applied element-wise. Scaling with the square root of the width is classical, although it often appears as an equivalent condition on the variance at initialization (Glorot & Bengio, 2010; LeCun et al., 2012; He et al., 2015). We make the scaling factors explicit to have weights of magnitude $\mathcal{O}(1)$ independently of the width and the depth. The $1/L$ scaling factor is less common, but it is necessary for the correspondence with neural ODEs to hold. More precisely, if there exist Lipschitz continuous functions $\mathcal{V}$ and $\mathcal{W}$ such that $V_k = \mathcal{V}(k/L)$ and $W_k = \mathcal{W}(k/L)$, then the residual network (1) converges, as $L \to \infty$, to the ODE

$$\frac{dH}{ds}(s) = \frac{1}{\sqrt{m}} \mathcal{V}(s) \sigma\Big(\frac{1}{\sqrt{q}} \mathcal{W}(s) H(s)\Big), \quad s \in [0, 1], \tag{2}$$

---

*Equal contribution. Correspondence to `pierre.marion@mines.org`.

where $s$ is the continuous-depth version of the layer index. It is important to note that this correspondence holds for *fixed* limiting functions $\mathcal{V}$ and $\mathcal{W}$. This is especially true at initialization, for example by setting the $V_k$ to zero and the $W_k$ to Gaussian matrices weight-tied across the depth. The initial residual network is then trivially equal to the neural ODE $\frac{dH}{ds}(s) = 0$. Of course, more sophisticated initializations are possible, as shown, e.g., in Marion et al. (2022); Sander et al. (2022b). However, regardless of an ODE structure at initialization, a more challenging question is that of the structure of the network *during and after* training. Since the weights are updated during training, there is a priori no guarantee that an ODE limit still holds after training, even if it does at initialization.

The question of a potential ODE structure for the trained network is not a mere technical one. In fact, it is important for at least three reasons. First, it gives a precise answer to the question of the connection between (trained) residual networks and neural ODEs, providing more solid ground to a common statement in the community that both can coincide in the large-depth limit (see, e.g., Haber & Ruthotto, 2017; E et al., 2019; Dong et al., 2020; Massaroli et al., 2020; Kidger, 2022). Second, it opens exciting perspectives for understanding residual networks. Indeed, if trained residual networks are discretizations of neural ODEs, then it is possible to apply results from neural ODEs to the large family of residual networks. In particular, from a theoretical point of view, the approximation capabilities of neural ODEs are well understood (Teshima et al., 2020; Zhang et al., 2020) and it is relatively easy to obtain generalization bounds for these models (Hanson & Raginsky, 2022; Marion, 2023). From a practical standpoint, advantages of neural ODEs include memory-efficient training (Chen et al., 2018; Sander et al., 2022b) and weight compression (Queiruga et al., 2021). This is important because in practice memory is a bottleneck for training residual networks (Gomez et al., 2017). Finally, our analysis is a first step towards understanding the implicit regularization (Neyshabur et al., 2014; Vardi, 2023) of gradient descent for deep residual networks, that is, characterizing the properties of the trained network among all minimizers of the empirical risk.

Throughout the document, it is assumed that the network is trained with gradient flow, which is a continuous analog of gradient descent. The parameters $V_k$ are updated according to an ODE of the form $\frac{dV_k}{dt}(t) = -L \frac{\partial \ell}{\partial V_k}(t)$ for $t \geqslant 0$, where $\ell$ is an empirical risk (the exact mathematical context and assumptions are detailed in Section 3), and similarly for $W_k$. The scaling factor $L$ is the counterpart of the factor $1/L$ in (1), and prevents vanishing gradients as $L$ tends to infinity. Note that the gradient flow is defined with respect to a time index $t$ different from the layer index $s$.

**Contributions.** Our first main contribution (Section 4.1) is to show that a neural ODE limit holds after training up to time $t$, i.e., there exists a function $\mathcal{V}(s, t)$ such that the residual network converges, as $L$ tends to infinity, to the ODE

$$\frac{dH}{ds}(s) = \frac{1}{\sqrt{m}} \mathcal{V}(s,t) \sigma \left( \frac{1}{\sqrt{q}} \mathcal{W}(s,t) H(s) \right), \quad s \in [0,1].$$

This large-depth limit holds for any finite training time $t \geqslant 0$. However, the convergence of the optimization algorithm as $t$ tends to infinity, which we refer to as the *long-time limit* to distinguish it from the large-depth limit $L \to \infty$, is not guaranteed without further assumptions, due to the non-convexity of the optimization problem. We attack the question (Section 4.2) when the width is large enough by proving a Polyak-Łojasiewicz (PL) condition, which is now state of the art in analyzing the properties of optimization algorithms for deep neural networks (Liu et al., 2022). The main assumption for our PL condition to hold is that the width $m$ of the hidden layers should be greater than some constant times the number of data $n$. As a second main contribution, we show that the PL condition yields the long-time convergence of the gradient flow for residual networks with linear overparameterization. Finally, we prove the convergence with high probability in the long-time limit, namely the existence of functions $\mathcal{V}_\infty$ and $\mathcal{W}_\infty$ such that the discrete trajectory defined by the trained residual network (1) converges as *both* $L$ and $t$ tend to infinity to the solution of the neural ODE (2) with $\mathcal{V} = \mathcal{V}_\infty$ and $\mathcal{W} = \mathcal{W}_\infty$. In addition, our approach points out that this limiting ODE interpolates the training data. Finally, our results are illustrated by numerical experiments (Section 5).

## 2 RELATED WORK

**Deep residual networks and neural ODEs.** Several works study the large-depth convergence of residual networks to differential equations, but without considering the training dynamics (Thorpe & van Gennip, 2023; Cohen et al., 2021; Marion et al., 2022; Hayou, 2023). Closer to our setting, Cont

et al. (2022) and Sander et al. (2022b) analyze the dynamics of gradient descent for deep residual networks, as we do, but with significant differences. Cont et al. (2022) consider a $1/\sqrt{L}$ scaling factor in front of the residual branch, resulting in a limit that is not a neural ODE. In addition, only $W$ is trained. Furthermore, to obtain convergence in the long-time limit, it is assumed that the data points are nearly orthogonal. Sander et al. (2022b) prove the existence of an ODE limit for trained residual networks, but in the simplified case of a linear activation and under a more restricted setting.

**Long-time convergence of wide residual networks.** Polyak-Łojasiewicz conditions are a modern tool to prove long-time convergence of overparameterized neural networks (Liu et al., 2022). These conditions are a relaxation of convexity, and mean that the gradients of the loss with respect to the parameters cannot be small when the loss is large. They have been applied to residual networks with both linear (Bartlett et al., 2018; Wu et al., 2019; Zou et al., 2020) and nonlinear activations (Allen-Zhu et al., 2019; Frei et al., 2019; Barboni et al., 2022; Cont et al., 2022; MacDonald et al., 2022). Building on the proof technique of Nguyen & Mondelli (2020) for non-residual networks, we need only a linear overparameterization to prove our PL condition, i.e., we require $m = \Omega(n)$. This compares favorably with results requiring polynomial overparameterization (Allen-Zhu et al., 2019; Barboni et al., 2022) or assumptions on the data, either a margin condition (Frei et al., 2019) or a sample size smaller than the dimension of the data space (Cont et al., 2022; MacDonald et al., 2022).

**Implicit regularization.** Our paper can be related to a line of work on the implicit regularization of gradient-based algorithms for residual networks (Neyshabur et al., 2014). We show that the optimization algorithm does not just converge to any residual network that minimizes the empirical risk, but rather to the discretization of a neural ODE. Note that most implicit regularization results state that the optimization algorithm converges to an interpolator that minimizes some complexity measure, which can be a margin (Lyu & Li, 2020), a norm (Boursier et al., 2022), or a matrix rank (Li et al., 2021). Thus, an interesting next step is to understand if the neural ODE found by gradient flow actually minimizes some complexity measure, and to characterize its generalization properties.

## 3 DEFINITIONS AND NOTATION

This section is devoted to specifying the setup outlined in Section 1. Proofs are given in the appendix.

**Residual network.** A (scaled) residual network of depth $L \in \mathbb{N}^*$ is defined by

$$h_0^L = A^L x$$
$$h_{k+1}^L = h_k^L + \frac{1}{L\sqrt{m}} V_{k+1}^L \sigma\Big(\frac{1}{\sqrt{q}} W_{k+1}^L h_k^L\Big), \quad k \in \{0, \dots, L-1\}, \tag{3}$$
$$F^L(x) = B^L h_L^L.$$

To allow the hidden layers $h_k^L \in \mathbb{R}^q$ to have a different dimension than the input $x \in \mathbb{R}^d$, we first map $x$ to $h_0^L$ with a weight matrix $A^L \in \mathbb{R}^{q \times d}$. We assume that the hidden layers belong to a higher dimensional space than the input and output, i.e., $q \geqslant \max(d, d')$. The residual transformations are two-layer perceptrons parameterized by the weight matrices $V_k^L \in \mathbb{R}^{q \times m}$ and $W_k^L \in \mathbb{R}^{m \times q}$. This is standard in the literature (e.g., He et al., 2016a; Dupont et al., 2019; Barboni et al., 2022). The last weight matrix $B^L \in \mathbb{R}^{d' \times q}$ maps the last hidden layer to the output $F^L(x)$ in $\mathbb{R}^{d'}$. Also, $\sigma : \mathbb{R} \to \mathbb{R}$ is an element-wise activation function assumed to be $\mathcal{C}^2$, non-constant, Lipschitz continuous, bounded, and such that $\sigma(0) = 0$. The convenient shorthand $Z_k^L = (V_k^L, W_k^L)$ is occasionally used, and we denote $\|Z_k^L\|_F$ the sum of the Frobenius norms $\|V_k^L\|_F + \|W_k^L\|_F$.

**Data and loss.** The data is a sample of $n$ pairs $(x_i, y_i)_{1 \leqslant i \leqslant n} \in (\mathcal{X} \times \mathcal{Y})^n$ where $\mathcal{X} \times \mathcal{Y}$ is a compact set of $\mathbb{R}^d \times \mathbb{R}^{d'}$. The empirical risk is the mean squared error $\ell^L = \frac{1}{n} \sum_{i=1}^n \|F^L(x_i) - y_i\|^2$.

**Initialization.** We initialize $A^L = (I_{\mathbb{R}^{d \times d}}, 0_{\mathbb{R}^{(q-d) \times d}})$ as the identity matrix in $\mathbb{R}^{d \times d}$ concatenated row-wise with the zero matrix in $\mathbb{R}^{(q-d) \times d}$, to act as a simple projection of the input onto the higher dimensional space $\mathbb{R}^q$, and similarly $B^L = (0_{\mathbb{R}^{d' \times (q-d')}}, I_{\mathbb{R}^{d' \times d'}})$. The weights $V_k^L$ are initialized to zero and the $W_k^L$ as weight-tied standard Gaussian matrices, i.e., for all $k \in \{1, \dots, L\}$,

$W_k^L = W \sim \mathcal{N}(0,1)^{\otimes(m \times q)}$. Initializing outer matrices to zero is standard practice (Zhang et al., 2019), while taking weight-tied matrices instead of i.i.d. ones is less common. We show in Section 5 that it is still possible to learn with this initialization scheme on real world data. As explained in Section 4.3, other initialization choices are possible, provided they correspond to the discretization of a Lipschitz continuous function, but we focus on this one in the main text for simplicity.

**Training algorithm.** Gradient flow is the limit of gradient descent as the learning rate tends to zero. The parameters are set at time $t = 0$ by the initialization, and then evolve according to the ODE

$$\frac{dA^L}{dt}(t) = -\frac{\partial \ell^L}{\partial A^L}(t), \quad \frac{dZ_k^L}{dt}(t) = -L\frac{\partial \ell^L}{\partial Z_k^L}(t), \quad \frac{dB^L}{dt}(t) = -\frac{\partial \ell^L}{\partial B^L}(t), \quad t \geqslant 0, \quad (4)$$

for $k \in \{1, \ldots, L\}$. In the following, the dependence in $t$ is made explicit when necessary, e.g., we write $h_k^L(t)$ instead of $h_k^L$, and $F^L(x;t)$ instead of $F^L(x)$.

It turns out that, without further assumptions, the gradient flow can diverge in finite time, because the dynamics are not (globally) Lipschitz continuous. A common practice (Goodfellow et al., 2016, Section 10.11.1) is to consider instead a clipped gradient flow

$$\frac{dA^L}{dt}(t) = \pi\Big(-\frac{\partial \ell^L}{\partial A^L}(t)\Big), \quad \frac{dZ_k^L}{dt}(t) = \pi\Big(-L\frac{\partial \ell^L}{\partial Z_k^L}(t)\Big), \quad \frac{dB^L}{dt}(t) = \pi\Big(-\frac{\partial \ell^L}{\partial B^L}(t)\Big), \quad (5)$$

where $\pi$ is a generic notation for a bounded Lipschitz continuous operator. For example, clipping each coordinate of the gradient at some $C > 0$ amounts to taking $\pi$ as the projection on the ball centered at 0 of radius $C$ for the $\ell_\infty$ norm. Clipping ensures that the gradient flow does not diverge, hence the dynamics are well defined, as a consequence of the Picard-Lindelöf theorem (see Lemma 16).

**Proposition 1.** *The (clipped) gradient flow* (5) *has a unique solution for all* $t \geqslant 0$.

In Section 4.2, we make additional assumptions to prove the long-time convergence of the gradient flow. We then prove that these assumptions ensure that the dynamics of the gradient flow (4) are well defined, eliminating the need for clipping (since in this case we show that the gradients are bounded).

**Neural ODE.** The neural ODE corresponding to the residual network (3) is defined by

$$\begin{aligned} H(0) &= Ax \\ \frac{dH}{ds}(s) &= \frac{1}{\sqrt{m}}\mathcal{V}(s)\sigma\Big(\frac{1}{\sqrt{q}}\mathcal{W}(s)H(s)\Big), \quad s \in [0,1], \\ F(x) &= BH(1), \end{aligned} \quad (6)$$

where $x \in \mathbb{R}^d$ is the input, $H \in \mathbb{R}^q$ is the variable of the ODE, $\mathcal{V} : [0,1] \to \mathbb{R}^{q \times m}$ and $\mathcal{W} : [0,1] \to \mathbb{R}^{m \times q}$ are Lipschitz continuous functions, $A \in \mathbb{R}^{q \times d}$ and $B \in \mathbb{R}^{d' \times q}$ are matrices, and the output is $F(x) \in \mathbb{R}^{d'}$. The following proposition shows that the neural ODE is well defined. In addition, its output is close to the residual network (3) provided the weights are discretizations of $\mathcal{V}$ and $\mathcal{W}$.

**Proposition 2.** *The neural ODE* (6) *has a unique solution* $H : [0,1] \to \mathbb{R}^q$. *Consider, moreover, the residual network* (3) *with* $A^L = A$, $V_k^L = \mathcal{V}(k/L)$ *and* $W_k^L = \mathcal{W}(k/L)$ *for* $k \in \{1, \ldots, L\}$, *and* $B^L = B$. *Then there exists* $C > 0$ *such that, for all* $L \in \mathbb{N}^*$, $\sup_{x \in \mathcal{X}} \|F(x) - F^L(x)\| \leqslant \frac{C}{L}$.

Clearly, our choices of $V_k^L$ and $W_k^L$ at initialization are discretizations of the Lipschitz continuous (in fact, constant) functions $\mathcal{V}(s) \equiv 0$ and $\mathcal{W}(s) \equiv W \sim \mathcal{N}(0,1)^{\otimes(m \times q)}$. Thus, Proposition 2 holds at initialization, and the residual network is equivalent to the trivial ODE $\frac{dH}{ds}(s) = 0$. The next section shows that after training we obtain non-trivial dynamics, which still discretize neural ODEs.

## 4 LARGE-DEPTH LIMIT OF RESIDUAL NETWORKS

We study the large-depth limit of trained residual networks in two settings. In Section 4.1, we consider the case of a finite training time. We move in Section 4.2 to the case where the training time tends to infinity, which is tractable under a Polyak-Łojasiewicz condition. Proofs are given in the appendix.

### 4.1 CLIPPED GRADIENT FLOW AND FINITE TRAINING TIME

We first consider the case where the neural network is trained with clipped gradient flow (5) on some training time interval $[0, T]$, $T > 0$. This allows us to prove large-depth convergence to a neural ODE without further assumptions. We emphasize that stopping training after a finite training time is a common technique in practice, referred to as early stopping (Goodfellow et al., 2016, Section 7.8). It is considered as a form of implicit regularization, and our result sheds light on this intuition by showing that the complexity of the trained networks increases with $T$.

The following proposition is a key step in proving the main theorem of this section.

**Proposition 3.** *There exist $M, K > 0$ such that, for any $t \in [0, T]$, $L \in \mathbb{N}^*$, and $k \in \{1, \ldots, L\}$,*

$$\max \left( \left\| A^L(t) \right\|_F, \left\| V_k^L(t) \right\|_F, \left\| W_k^L(t) \right\|_F, \left\| B^L(t) \right\|_F \right) \leqslant M,$$

*and, for $k \in \{1, \ldots, L-1\}$,*

$$\max \left( \left\| V_{k+1}^L(t) - V_k^L(t) \right\|_F, \left\| W_{k+1}^L(t) - W_k^L(t) \right\|_F \right) \leqslant \frac{K}{L}.$$

*Moreover, with probability at least $1 - \exp \left( - \frac{3qm}{16} \right)$, the following expressions hold for $M$ and $K$:*

$$M = TM_\pi + 2\sqrt{qm}, \quad K = \beta T e^{\alpha T}, \tag{7}$$

*where $M_\pi$ is the supremum of $\pi$ in Frobenius norm, and $\alpha$ and $\beta$ depend on $\mathcal{X}$, $\mathcal{Y}$, $M$, and $\sigma$.*

This proposition ensures that the size of the weights and the difference between successive weights remain bounded throughout training. We can now state the main result, which states the convergence, for any training time in $[0, T]$, of the neural network to a neural ODE as $L \to \infty$. Recall that a sequence of functions $f^L$ converges uniformly over $u \in U$ to $f$ if $\sup_{u \in U} \|f^L(u) - f(u)\| \to 0$.

**Theorem 4.** *Consider the residual network (3) with the training dynamics (5). Then the following statements hold **as $L$ tends to infinity**:*

(i) *There exist functions $A : [0, T] \to \mathbb{R}^{q \times d}$ and $B : [0, T] \to \mathbb{R}^{d' \times q}$ such that $A^L(t)$ and $B^L(t)$ converge uniformly over $t \in [0, T]$ to $A(t)$ and $B(t)$.*

(ii) *There exists a Lipschitz continuous function $\mathcal{Z} : [0, 1] \times [0, T] \to \mathbb{R}^{q \times m} \times \mathbb{R}^{m \times q}$ such that*

$$\mathcal{Z}^L : [0, 1] \times [0, T] \to \mathbb{R}^{q \times m} \times \mathbb{R}^{m \times q}, \ (s, t) \mapsto \mathcal{Z}^L(s, t) = Z^L_{\lfloor (L-1)s \rfloor + 1}(t) \tag{8}$$

*converges uniformly over $s \in [0, 1]$ and $t \in [0, T]$ to $\mathcal{Z} := (\mathcal{V}, \mathcal{W})$.*

(iii) *Uniformly over $s \in [0, 1]$, $t \in [0, T]$, and $x \in \mathcal{X}$, the hidden layer $h^L_{\lfloor Ls \rfloor}(t)$ converges to the solution at time $s$ of the neural ODE*

$$\begin{aligned} H(0, t) &= A(t)x \\ \frac{\partial H}{\partial s}(s, t) &= \frac{1}{\sqrt{m}} \mathcal{V}(s, t) \sigma \left( \frac{1}{\sqrt{q}} \mathcal{W}(s, t) H(s, t) \right), \quad s \in [0, 1]. \end{aligned} \tag{9}$$

(iv) *Uniformly over $t \in [0, T]$ and $x \in \mathcal{X}$, the output $F^L(x; t)$ converges to $B(t)H(1, t)$.*

Let us sketch the proof of statement $(ii)$, which is the cornerstone of the theorem. A first key idea is to introduce in (8) the piecewise-constant continuous-depth interpolation $\mathcal{Z}^L$ of the weights, whose ambient space does not depend on $L$, in contrast to the discrete weight sequence $Z_k^L$. Since the weights remain bounded during training by Proposition 3, the Arzelà-Ascoli theorem guarantees the existence of an accumulation point for $\mathcal{Z}^L$. We show that the accumulation point is unique because it is the solution of an ODE satisfying the conditions of the Picard-Lindelöf theorem. The uniqueness of the accumulation point then implies the existence of a limit for the weights.

There are two notable byproducts of our proof. The first one is an explicit description of the training dynamics of the limiting weights $A$, $B$, and $\mathcal{Z}$, as the solution of an ODE system, as presented in Appendix A.5. The second one, which we now describe, consists of norm bounds on the weights. Proposition 3 bounds the discrete weights and the difference between two consecutive weights

respectively by some $M, K > 0$. The proof of Theorem 4 shows that this bound carries over to the continuous weights, in the sense that $A(t)$, $\mathcal{V}(s,t)$, $\mathcal{W}(s,t)$, and $B(t)$ are uniformly bounded by $M$, and $\mathcal{V}(\cdot, t)$ and $\mathcal{W}(\cdot, t)$ are uniformly Lipschitz continuous with Lipschitz constant $K$. Formally, this last property means that, for any $s, s' \in [0,1]$ and $t \in [0, T]$,

$$\|\mathcal{V}(s', t) - \mathcal{V}(s,t)\|_F \leqslant K|s' - s| \quad \text{and} \quad \|\mathcal{W}(s', t) - \mathcal{W}(s,t)\|_F \leqslant K|s' - s|.$$

A key point to obtain this result is that $K$ and $M$ in Proposition 3 are independent of $L$. This would not be the case if we had naively bounded in Proposition 3 the difference between two successive weight matrices by a constant, without taking into account the smoothness of the weights. The boundedness and Lipschitz continuity of the weights are important features because they limit the statistical complexity of neural ODEs (Marion, 2023). More generally, norm-based bounds are a common approach in the statistical theory of deep learning (see, e.g., Bartlett et al., 2017, and references therein). Looking at the formula (7) for $M$ and $K$, one can see in particular that the bounds diverge exponentially with $T$, providing an argument in favor of early stopping.

Our approach so far characterizes the large-depth limit of the neural network for a finite training time $T$, but two questions remain open. A first challenge is to characterize the value of the loss after training. A second one is to provide insight into the convergence of the optimization algorithm in the long-time limit, i.e., as $T$ tends to infinity. To answer these questions, we move to the setting where the width of the network is large enough, which allows us to prove a Polyak-Łojasiewicz (PL) condition and thereby the long-time convergence of the training loss to zero.

## 4.2 Convergence in the long-time limit for wide networks

We now introduce the definition (with the notation $Z^L = (V_k^L, W_k^L)_{k \in \{1, \ldots, L\}}$) of the PL condition:

**Definition 1.** *For $M, \mu > 0$, the residual network (3) is said to satisfy the $(M, \mu)$-local PL condition around a set of parameters $(\bar{A}^L, \bar{Z}^L, \bar{B}^L)$ if, for every set of parameters $(A^L, Z^L, B^L)$ such that*

$$\|A^L - \bar{A}^L\|_F \leqslant M, \quad \sup_{k \in \{1, \ldots, L\}} \|Z_k^L - \bar{Z}_k^L\|_F \leqslant M, \quad \|B^L - \bar{B}^L\|_F \leqslant M,$$

*one has*

$$\left\|\frac{\partial \ell^L}{\partial A^L}\right\|_F^2 + L \sum_{k=1}^L \left\|\frac{\partial \ell^L}{\partial Z_k^L}\right\|_F^2 + \left\|\frac{\partial \ell^L}{\partial B^L}\right\|_F^2 \geqslant \mu \ell^L,$$

*where the loss $\ell^L$ is evaluated at the set of parameters $(A^L, Z^L, B^L)$.*

The next important point is to observe that, under the setup of Section 3 and some additional assumptions, the residual network satisfies the local PL condition of Definition 1.

**Proposition 5.** *Assume that the sample points $(x_i, y_i)$ are i.i.d. such that $\|x_i\|_2 = \sqrt{q}$. Then there exist $c_1, \ldots, c_4 > 0$ (depending only on $\sigma$) and $\delta > 0$ such that, if*

$$q \geqslant d + d', \quad m \geqslant c_1 n, \quad L \geqslant c_2 \sqrt{nq},$$

*then, with probability at least $1 - \delta$, the residual network (3) satisfies the $(M, \mu)$-local PL condition around its initialization, with $M = c_3 / \sqrt{nq}$ and $\mu = c_4 / (n\sqrt{nq})$.*

We emphasize that Proposition 5 requires the width $m$ to scale only linearly with the sample size $n$, which improves on the literature (see Section 2). The other assumptions are mild. Note that our proof shows that the parameter $\delta$ is small if $n$ grows at most polynomially with $d$ (see Appendix B.5).

We are now ready to state convergence in the long-time and large-depth limits to a global minimum of the empirical risk, when the local PL condition holds and the norm of the targets $y_i$ is small enough.

**Theorem 6.** *Consider the residual network (3) with the training dynamics (4), and assume that the assumptions of Proposition 5 hold. Then there exist $C, \delta > 0$ such that, if $\frac{1}{n} \sum_{i=1}^n \|y_i\|^2 \leqslant C$, then, with probability as least $1 - \delta$, the gradient flow is well defined on $\mathbb{R}_+$, and, for $t \in \mathbb{R}_+$ and $L \in \mathbb{N}^*$,*

$$\ell^L(t) \leqslant \exp\left(-\frac{C't}{n\sqrt{nq}}\right)\ell^L(0), \tag{10}$$

*for some $C' > 0$ depending on $\sigma$. Moreover, the following statements hold as $t$ **and** $L$ **tend to infinity**:*

(i) *There exist matrices $A_\infty \in \mathbb{R}^{q \times d}$ and $B_\infty \in \mathbb{R}^{d' \times q}$ such that $A^L(t)$ and $B^L(t)$ converge to $A_\infty$ and $B_\infty$.*

(ii) *There exists a Lipschitz continuous function $\mathcal{Z}_\infty : [0,1] \to \mathbb{R}^{q \times m} \times \mathbb{R}^{m \times q}$ such that*
$$\mathcal{Z}^L : [0,1] \times \mathbb{R}_+ \to \mathbb{R}^{q \times m} \times \mathbb{R}^{m \times q}, \quad (s,t) \mapsto \mathcal{Z}^L(s,t) = Z^L_{\lfloor (L-1)s \rfloor + 1}(t)$$
*converges uniformly over $s \in [0,1]$ to $\mathcal{Z}_\infty := (\mathcal{V}_\infty, \mathcal{W}_\infty)$.*

(iii) *Uniformly over $s \in [0,1]$ and $x \in \mathcal{X}$, the hidden layer $h^L_{\lfloor Ls \rfloor}(t)$ converges to the solution at time $s$ of the neural ODE*
$$H(0) = A_\infty x$$
$$\frac{dH}{ds}(s) = \frac{1}{\sqrt{m}} \mathcal{V}_\infty(s) \sigma\Big(\frac{1}{\sqrt{q}} \mathcal{W}_\infty(s) H(s)\Big), \quad s \in [0,1].$$

(iv) *Uniformly over $x \in \mathcal{X}$, the output $F^L(x;t)$ converges to $F_\infty(x) = B_\infty H(1)$. Furthermore, $F_\infty(x_i) = y_i$ for all $i \in \{1, \ldots, n\}$.*

This theorem proves two important results of separate interest. On the one hand, equation (10) shows the long-time convergence of the gradient flow for deep residual networks under the linear overparameterization assumption $m \geqslant c_1 n$ of Proposition 5. On the other hand, when both $t$ and $L$ tend to infinity, the network converges to a neural ODE that further interpolates the training data. Note that the order in which $t$ and $L$ tend to infinity does not matter by uniform convergence properties.

## 4.3 GENERALIZATIONS TO OTHER ARCHITECTURES AND INITIALIZATION

To simplify the exposition, we have so far considered a particular residual architecture defined in (3). However, most of our results hold for a more general residual network of the form

$$h^L_{k+1} = h^L_k + \frac{1}{L} f(h^L_k, Z^L_{k+1}), \quad k \in \{0, \ldots, L-1\}, \tag{11}$$

where $f : \mathbb{R}^q \times \mathbb{R}^p \to \mathbb{R}^q$ is a $\mathcal{C}^2$ function such that $f(0, \cdot) \equiv 0$ and $f(\cdot, z)$ is uniformly Lipschitz for $z$ in any compact. All our results are shown in the appendix for this general model, except the PL condition of Proposition 5, which we prove only for the specific setup of Section 3. In particular, the conclusions of Theorem 4 hold for the general model (11), as well as those of Theorem 6 if the network satisfies a $(M, \mu)$-local PL condition with $\mu$ sufficiently large (see Appendix B for details).

Our network of interest (3) is a special case of model (11), and other choices include convolutional layers (or any sparse version of (3)) or a Lipschitz continuous version of Transformer (Kim et al., 2021). This latter case is particularly interesting in the light of the literature analyzing Transformer from a neural ODE point of view (Lu et al., 2019; Sander et al., 2022a; Geshkovski et al., 2023).

Moreover, the initialization assumption made in Section 3 can also be relaxed to include any so-called *smooth* initialization of the weights (Marion et al., 2022). A smooth initialization corresponds to taking $V^L_k(0)$ and $W^L_k(0)$ as discretizations of some Lipschitz continuous functions $\mathcal{V}_0 : [0,1] \to \mathbb{R}^{q \times m}$ and $\mathcal{W}_0 : [0,1] \to \mathbb{R}^{m \times q}$, that is, for $k \in \{1, \ldots, L\}$, $V^L_k(0) = \mathcal{V}_0(\frac{k}{L})$ and $W^L_k(0) = \mathcal{W}_0(\frac{k}{L})$. A typical concrete example is to let the entries of $\mathcal{V}_0$ and $\mathcal{W}_0$ be independent Gaussian processes with expectation zero and squared exponential covariance $K(x, x') = \exp(-\frac{(x-x')^2}{2\ell^2})$, for some $\ell > 0$. As shown by Proposition 2, a smooth initialization means that the network discretizes a neural ODE.

## 5 NUMERICAL EXPERIMENTS

We now present numerical experiments to validate our theoretical findings, using both synthetic and real-world data. Our code is available on GitHub (see Appendix E for details and additional plot).

## 5.1 SYNTHETIC DATA

We consider the residual network (3) with the initialization scheme of Section 3. The activation function is GELU (Hendrycks & Gimpel, 2016), which is a smooth approximation of ReLU: $x \mapsto$

$\max(x, 0)$. The sample points $(x_i, y_i)_{1 \leqslant i \leqslant n}$ follow independent standard Gaussian distributions. The mean-squared error is minimized using full-batch gradient descent. The following experiments exemplify the large-depth ($t \in [0, T]$, $L \to \infty$) and long-time ($t \to \infty$, $L$ finite) limits.

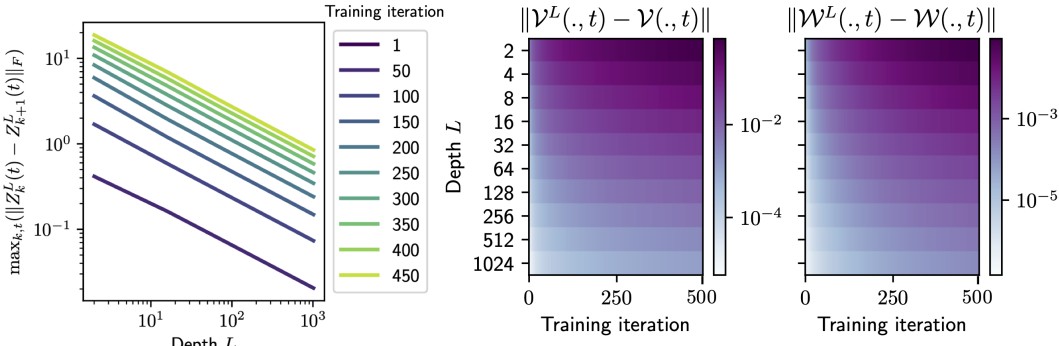

Figure 1: **Left**: $1/L$ convergence of the maximum distance between two successive weight matrices $\max_{1 \leqslant k \leqslant L, t \in [0,T]}(\|Z_k^L(t) - Z_{k+1}^L(t)\|_F)$. **Right**: uniform convergence of $\mathcal{Z}^L$ to its large-depth limit $\mathcal{Z}$. Here, for a matrix-valued function $f$, $\|f\|$ denotes $(\int_0^1 \|f(s)\|_F^2 ds)^{1/2}$.

**Large-depth limit.** We illustrate key insights of Proposition 3 and Theorem 4, with $T = 500$. In Figure 1 (left), we plot the maximum distance between two successive weight matrices, i.e., $\max_{1 \leqslant k \leqslant L, t \in [0,T]}(\|Z_k^L(t) - Z_{k+1}^L(t)\|_F)$, for different values of $L$ and training time $T$. We observe a $1/L$ convergence rate, as predicted by Proposition 3. Moreover, for a fixed $L$, the distance between two successive weight matrices increases with the training time, however at a much slower pace than the exponential upper bound on $K$ given in identity (7). Figure 1 (right) depicts the uniform convergence of $\mathcal{Z}^L$ to its large-depth limit $\mathcal{Z}$, illustrating statement $(ii)$ of Theorem 4. The function $\mathcal{Z}$ is computed using $\mathcal{Z}^L$ for $L = 2^{14}$. Note that the convergence is slower for larger training times.

**Long-time limit.** We now turn to the long-time training setup, training for 80,000 iterations with $L = 64$ and $m$ large enough to satisfy the assumptions of Theorem 6. In Figure 2, we plot a specific (randomly-chosen) entry of matrices $V_k^L$ and $W_k^L$ across layers, for different training times. This illustrates Theorem 6 in a practical setting since, visually, the weights behave as a Lipschitz continuous function for any training time and converge to a Lipschitz continuous function as $t \to \infty$. The loss decays to zero as a function of training time, also corroborating Theorem 6.

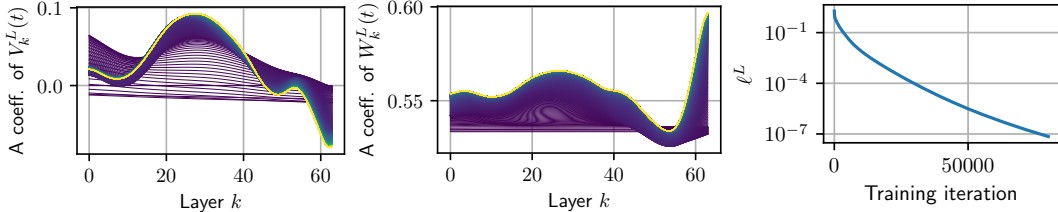

Figure 2: **Left**: Randomly-chosen entry of the weight matrices across layers ($x$-axis) for various training times $t$ (lighter color indicates higher training time). **Right**: Loss against training time.

## 5.2 REAL-WORLD DATA

We now investigate the properties of deep residual networks on the CIFAR 10 dataset (Krizhevsky, 2009). We deviate from the mathematical model (3) by using convolutions instead of fully connected layers. More precisely, $A^L$ is replaced by a trainable convolutional layer, and the residual layers write $h_{k+1}^L = h_k^L + \frac{1}{L}\text{bn}_{2,k}^L(\text{conv}_{2,k}^L(\sigma(\text{bn}_{1,k}^L(\text{conv}_{1,k}^L(h_k^L)))))$, where $\text{conv}_{i,k}^L$ are convolutions and $\text{bn}_{i,k}^L$ are batch normalizations (see Appendix E for discussion about normalization). The output of the residual layers is mapped to logits through a linear layer $B^L$. We initialize $\text{bn}_{2,k}^L$ to 0, and $\text{bn}_{1,k}^L$

and $\mathrm{conv}_{i,k}^L$ either to weight-tied or to i.i.d. Gaussian. Table 1 reports the accuracy of the trained network, and whether it has Lipschitz continuous (or smooth) weights after training, depending on the activation function $\sigma$ and on the initialization scheme. To assess the smoothness of the weights, we simply resort to visual inspection. For example, Figure 3 (left) shows two random entries of the convolutions across layers with GELU and a weight-tied initialization: the smoothness is preserved after training. Smooth weights indicate that the residual network discretizes a neural ODE (see, e.g., Proposition 2). On the contrary, if an i.i.d. initialization is used, smoothness is not preserved after training, as shown in Figure 3 (right), and the residual network does not discretize a neural ODE.

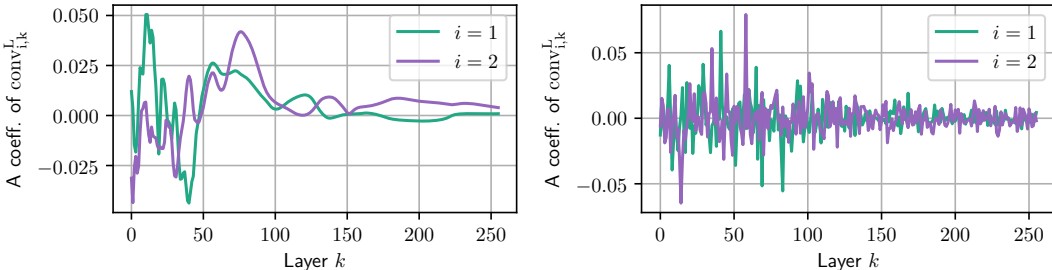

Figure 3: Random entries of the convolutions across layers ($x$-axis) after training. **Left:** Weight-tied initialization leads to smooth weights. **Right:** i.i.d. initialization leads to non-smooth weights.

| Act. function | Init. scheme | Train Acc. | Test Acc. | Smooth trained weights |
|---|---|---|---|---|
| Identity | Weight-tied | $56.5 \pm 0.1$ | $59.8 \pm 0.7$ | ✓ |
| | i.i.d. | $56.1 \pm 0.3$ | $59.6 \pm 0.7$ | ✗ |
| GELU | Weight-tied | $80.5 \pm 0.7$ | $79.9 \pm 0.2$ | ✓ |
| | i.i.d. | $89.8 \pm 0.5$ | $85.7 \pm 0.1$ | ✗ |
| ReLU | Weight-tied | $97.4 \pm 0.6$ | $88.1 \pm 0.1$ | ✗ |
| | i.i.d. | $98.4 \pm 0.1$ | $88.4 \pm 0.5$ | ✗ |

Table 1: Accuracy and smoothness of the trained weights depending on the choice of activation function $\sigma$ and initialization scheme. We display the median over 5 runs and the interquartile range between the first and third quantile. Smooth weights correspond to a neural ODE structure.

Table 1 conveys several important messages. First, in accordance with our theory (Theorem 4), we obtain a neural ODE structure when using a smooth activation function and weight-tied initialization (lines 1 and 3 of Table 1). This is not the case when using the non-smooth ReLU activation and/or i.i.d. initialization. In fact, we prove in Appendix D that the smoothness of the weights is lost when training with ReLU in a simple setting. Furthermore, the third line of Table 1 shows that it is possible to obtain a reasonable accuracy with a neural ODE structure, which, as emphasized in Section 1, also comes with theoretical and practical advantages. Nevertheless, we see an improvement in accuracy in cases corresponding to non-smooth weights, i.e., to a network that does *not* discretize an ODE.

## 6 CONCLUSION

We study the convergence of deep residual networks to neural ODEs. When properly scaled and initialized, residual networks trained with fixed-horizon gradient flow converge to neural ODEs as the depth tends to infinity. This result holds for very general architectures. In the case where both training time and depth tend to infinity, convergence holds under a local Polyak-Łojasiewicz condition. We prove such a condition for a family of deep residual networks with linear overparameterization.

The setting of neural ODE-like networks comes with strong guarantees, at the cost of some performance gap when compared with i.i.d. initialization as highlighted by the experimental section. Extending the mathematical large-depth study to i.i.d. instead of weight-tied initialization is an interesting problem for future research. Previous work suggests that the correct limit object is then a *stochastic* differential equation (Cohen et al., 2021; Cont et al., 2022; Marion et al., 2022).

ACKNOWLEDGMENTS

P.M. is supported by a grant from Région Île-de-France and by MINES Paris - PSL. P.M. and Y.-H.W. are funded by a Google PhD Fellowship. M.S. is supported by the "Investissements d'avenir" program, reference ANR19-P3IA-0001, and by the European Research Council (ERC project NORIA). This work was granted access to the HPC resources of IDRIS under the allocation 2020-[AD011012073] made by GENCI.

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

# Appendix

**Organization of the Appendix.** In Section A, we give some results on the general residual network (11). In Section B, these results are instantiated in the specific case of the residual network (3), thus proving the results of the paper. Section C contains some lemmas that are useful for the proofs. We present in Section D a counter-example showing that a residual network with the ReLU activation can move away from the neural ODE structure during training. Finally, Section E presents some experimental details.

## A    SOME RESULTS FOR GENERAL RESIDUAL NETWORKS

**Lipschitz continuity.**    Let $(\mathscr{U}, \|\cdot\|)$, $(\mathscr{V}, \|\cdot\|)$, and $(\mathscr{W}, \|\cdot\|)$ be generic normed spaces. Then a function of two variables $g : \mathscr{U} \times \mathscr{V} \to \mathscr{W}$ is:

(i) (Globally) Lipschitz continuous if there exists $K \geqslant 0$ such that, for $(u, v), (u', v') \in \mathscr{U} \times \mathscr{V}$,

$$\|g(u, v) - g(u', v')\| \leqslant K\|u - u'\| + K\|v - v'\|.$$

(ii) Locally Lipschitz continuous in its first variable if, for any compacts $E \subset \mathscr{U}, E' \subset \mathscr{V}$, there exists $K \geqslant 0$ such that, for $(u, v), (u', v) \in E \times E'$,

$$\|g(u, v) - g(u', v)\| \leqslant K\|u - u'\|.$$

Equivalent definitions hold for a function of one variable. Moreover, $g(\cdot, v)$ is said to be uniformly Lipschitz continuous for $v$ in $\mathscr{V}$ if there exists $K \geqslant 0$ such that, for $(u, v), (u', v) \in \mathscr{U} \times \mathscr{V}$,

$$\|g(u, v) - g(u', v)\| \leqslant K\|u - u'\|,$$

and uniformly Lipschitz continuous for $v$ in any compact if, for any compact $E' \subset \mathscr{V}$, there exists $K \geqslant 0$ such that, for $(u, v), (u', v) \in \mathcal{U} \times E'$,

$$\|g(u, v) - g(u', v)\| \leqslant K\|u - u'\|.$$

Throughout, we refer to a Lipschitz continuous function with Lipschitz constant $K \geqslant 0$ as $K$-Lipschitz.

**Model.**    As explained in Section 4.3, most of our results are proven for the general residual network

$$
\begin{aligned}
h_0^L(t) &= A^L(t)x \\
h_{k+1}^L(t) &= h_k^L(t) + \frac{1}{L}f(h_k^L(t), Z_{k+1}^L(t)), \quad k \in \{0, \dots, L-1\}, \\
F^L(x; t) &= B^L(t)h_L^L(t),
\end{aligned}
\tag{12}
$$

where $Z^L(t) = (Z_1^L(t), \dots, Z_L^L(t)) \in (\mathbb{R}^p)^L$ and $f : \mathbb{R}^q \times \mathbb{R}^p \to \mathbb{R}^q$ is a $\mathcal{C}^2$ function such that $f(0, \cdot) \equiv 0$ and $f(\cdot, z)$ is uniformly Lipschitz for $z$ in any compact. Let us introduce the backpropagation equations, which are instrumental in the study of the gradient flow dynamics. These equations define the backward state $p_k^L(t) = \frac{\partial \ell^L}{\partial h_k^L}(t) \in \mathbb{R}^q$ through the backward recurrence

$$
\begin{aligned}
p_L^L(t) &= 2B^L(t)^\top(F^L(x; t) - y) \\
p_k^L(t) &= p_{k+1}^L(t) + \frac{1}{L}\partial_1 f(h_k^L(t), Z_{k+1}^L(t))p_{k+1}^L(t), \quad k \in \{0, \dots, L-1\},
\end{aligned}
\tag{13}
$$

where $\partial_1 f \in \mathbb{R}^{q \times q}$ stands for the Jacobian matrix of $f$ with respect to its first argument. Similarly, we let $\partial_2 f \in \mathbb{R}^{q \times p}$ be the Jacobian matrix of $f$ with respect to its second argument. For a sample $(x_i, y_i)_{1 \leqslant i \leqslant n} \in (\mathcal{X} \times \mathcal{Y})^n$, we let $h_{k,i}^L(t)$ and $p_{k,i}^L(t)$ be, respectively, the hidden layer $h_k^L(t)$ and the backward state $p_k^L(t)$ associated with the $i$-th input $x_i$. Denoting the mean squared error associated

with the sample by $\ell^L$, we have, by the chain rule,

$$\frac{\partial \ell^L}{\partial A^L}(t) = \frac{1}{n} \sum_{i=1}^{n} p_{0,i}^L(t) x_i^\top \tag{14}$$

$$\frac{\partial \ell^L}{\partial Z_k^L}(t) = \frac{1}{nL} \sum_{i=1}^{n} \partial_2 f(h_{k-1,i}^L(t), Z_k^L(t))^\top p_{k,i}^L(t), \quad k \in \{1, \dots, L\}, \tag{15}$$

$$\frac{\partial \ell^L}{\partial B^L}(t) = \frac{2}{n} \sum_{i=1}^{n} (F^L(x_i; t) - y_i) h_{L,i}^L(t)^\top. \tag{16}$$

**Initialization.** The parameters $(Z_k^L(t))_{1 \leqslant k \leqslant L}$ are initialized to $Z_k^L(0) = Z^{\text{init}}\left(\frac{k}{L}\right)$, where $Z^{\text{init}} : [0,1] \to \mathbb{R}^p$ is a Lipschitz continuous function. Furthermore, we initialize $A^L(0)$ to some matrix $A^{\text{init}} \in \mathbb{R}^{q \times d}$ and $B^L(0) = B^{\text{init}} \in \mathbb{R}^{d' \times q}$. Note that this initialization scheme is a generalization of the one presented in Section 3.

**Additional notation.** For a vector $x$, $\|x\|$ denotes the Euclidean norm. For a matrix $A$, the operator norm induced by the Euclidean norm is denoted by $\|A\|_2$, and the Frobenius norm is denoted by $\|A\|_F$. Finally, we use the notation $A^L$ (resp. $Z_k^L$, $B^L$) to denote the function $t \mapsto A^L(t)$ (resp. $t \mapsto Z_k^L(t)$, $t \mapsto B^L(t)$), since the parameters are considered as functions of the training time throughout this appendix.

**Overview of Appendix A.** First, in Section A.1, we study the case of the (clipped) gradient flow (5). We show that the weights and the difference between successive weights are bounded during the entire training. Section A.2 shows a similar result for the standard gradient flow (4) under a PL condition. In Section A.3, we show a generalized version of the Arzelà-Ascoli theorem, which allows us to prove the existence of a converging subsequence of the weights in the large-depth limit. Section A.4 is devoted to the convergence of the Euler scheme for parameterized ODEs. We then proceed to prove in Section A.5 our main result, i.e., the large-depth convergence of the gradient flow. The key step is to establish the uniqueness of the adherence point of the weights. Finally, in Section A.6, we prove the existence of a double limit for the weights and the hidden states when both the depth and the training time tend to infinity.

## A.1 THE TRAINED WEIGHTS ARE BOUNDED IN THE FINITE TRAINING-TIME SETUP

Before stating the result, let us introduce the notation $\partial_{22} f(h, z) \in \mathbb{R}^{q \times p \times p}$, which is the third-order tensor of second partial derivatives of $f$ with respect to $z$. We endow the space $\mathbb{R}^{q \times p \times p}$ with the operator norm $\|\cdot\|_2$ induced by the Euclidean norm in $\mathbb{R}^p$ and the $\|\cdot\|_2$ norm in $\mathbb{R}^{q \times p}$. In other words,

$$\|\partial_{22} f(h, z)\|_2 = \sup_{u \in \mathbb{R}^p, \|u\|=1} \|\partial_{22} f(h, z) u\|_2,$$

where $\partial_{22} f(h, z) u \in \mathbb{R}^{q \times p}$ is the tensor product of $\partial_{22} f(h, z)$ against $u$. Similarly, $\partial_{21} f(h, z) \in \mathbb{R}^{q \times p \times q}$ denotes the third-order tensor of cross second partial derivatives of $f$, and the space $\mathbb{R}^{q \times p \times q}$ is endowed with the operator norm $\|\cdot\|_2$ induced by the Euclidean norm in $\mathbb{R}^q$ and the $\|\cdot\|_2$ norm in $\mathbb{R}^{q \times p}$.

**Proposition 7.** *Consider the residual network* (12) *initialized as explained in Appendix A and trained with the gradient flow* (5) *on* $[0, T]$, *for some* $T \in (0, \infty)$. *Let*

$$M_\pi = \max\left(\max_{A \in \mathbb{R}^{q \times d}} \|\pi(A)\|_F, \max_{Z \in \mathbb{R}^p} \|\pi(Z)\|, \max_{B \in \mathbb{R}^{d' \times q}} \|\pi(B)\|_F\right),$$

$$M_0 = \max\left(\|A^{\text{init}}\|_F, \sup_{s \in [0,1]} \|Z^{\text{init}}(s)\|, \|B^{\text{init}}\|_F\right) \quad \text{and} \quad M = M_0 + T M_\pi.$$

*Then the gradient flow is well defined on* $[0, T]$, *and, for* $t \in [0, T]$, $L \in \mathbb{N}^*$, *and* $k \in \{1, \dots, L\}$,

$$\|A^L(t)\|_F \leqslant M, \quad \|Z_k^L(t)\| \leqslant M, \quad \text{and} \quad \|B^L(t)\|_F \leqslant M. \tag{17}$$

*Moreover, there exist $\alpha, \beta > 0$ such that, for $t \in [0, T]$ and $k \in \{1, \dots, L-1\}$,*

$$\|Z_{k+1}^L(t) - Z_k^L(t)\| \leqslant \left(\|Z_{k+1}^L(0) - Z_k^L(0)\| + \frac{\beta T}{L}\right)e^{\alpha T}.$$

*The following expressions for $\alpha$ and $\beta$ hold:*

$\alpha = 2e^K K' M(e^K M^2 M_X + M_Y)$ *and* $\beta = 2Ke^K M(K + e^K K' M M_X)(e^K M^2 M_X + M_Y),$

*where*

$$M_X = \sup_{x \in \mathcal{X}} \|x\|, \quad M_Y = \sup_{y \in \mathcal{Y}} \|y\|, \quad K_1 = \sup_{\|z\| \leqslant M} \left\|\partial_1 f(h, z)\right\|_2 \tag{18}$$

$$E = \{(h, z) \in \mathbb{R}^d \times \mathbb{R}^p, \|h\| \leqslant e^{K_1} M M_X, \|z\| \leqslant M\} \tag{19}$$

$$K_2 = \sup_{(h,z) \in E} \left\|\partial_2 f(h, z)\right\|_2, \quad K = \max(K_1, K_2)$$

$$K' = \sup_{(h,z) \in E} \left(\max\left(\left\|\partial_{22} f(h, z)\right\|_2, \left\|\partial_{21} f(h, z)\right\|_2\right)\right).$$

*Proof.* The time-independent dynamics

$$(A^L, Z_k^L, B^L) \mapsto \left(\pi\left(-\frac{\partial \ell^L}{\partial A^L}\right), \pi\left(-L\frac{\partial \ell^L}{\partial Z_k^L}\right), \pi\left(-\frac{\partial \ell^L}{\partial B^L}\right)\right)$$

defining the gradient flow (5) are locally Lipschitz continuous, hence the gradient flow is defined on a maximal interval $[0, T_{\max})$ by the Picard-Lindelöf theorem (see Lemma 16). Let us show by contradiction that $T_{\max} = T$. Assume that $T_{\max} < T$. If this is true, again by the Picard-Lindelöf theorem, we know that the parameters diverge to infinity at $T_{\max}$. However, for any $t \in [0, T_{\max})$, we have

$$\|A^L(t)\|_F \leqslant \|A^L(0)\|_F + \int_0^t \left\|\frac{dA^L}{dt}(\tau)\right\|_F d\tau \leqslant M_0 + \int_0^t M_\pi d\tau \leqslant M_0 + T M_\pi = M.$$

Bounds on $B^L$ and $Z_k^L$ by $M$ can be shown similarly. This contradicts the divergence of the parameters at $t = T_{\max}$. We conclude that the gradient flow is well defined on $[0, T]$ and that the bounds (17) hold.

It remains to bound the difference $\|Z_{k+1}^L(t) - Z_k^L(t)\|$. We have, for $t \in [0, T]$ and $k \in \{1, \dots, L-1\}$,

$$\left\|\frac{dZ_{k+1}^L}{dt}(t) - \frac{dZ_k^L}{dt}(t)\right\| = L\left\|\frac{\partial \ell^L}{\partial Z_{k+1}^L}(t) - \frac{\partial \ell^L}{\partial Z_k^L}(t)\right\|$$

$$\leqslant \sum_{i=1}^n \frac{1}{n}\left\|\partial_2 f(h_{k,i}^L(t), Z_{k+1}^L(t))^\top p_{k,i}^L(t) - \partial_2 f(h_{k-1,i}^L(t), Z_k^L(t))^\top p_{k-1,i}^L(t)\right\|$$

$$\leqslant \frac{1}{n}\sum_{i=1}^n \left\|\partial_2 f(h_{k,i}^L(t), Z_{k+1}^L(t))\right\|_2 \left\|p_{k,i}^L(t) - p_{k-1,i}^L(t)\right\|$$

$$+ \left\|p_{k-1,i}^L(t)\right\| \left\|\partial_2 f(h_{k,i}^L(t), Z_{k+1}^L(t)) - \partial_2 f(h_{k-1,i}^L(t), Z_k^L(t))\right\|_2 \tag{20}$$

Furthermore, for $t \in [0, T]$, $k \in \{0, \dots, L-1\}$, and $i \in \{1, \dots, n\}$,

$$\|h_{k+1,i}^L(t)\| = \|h_{k,i}^L(t) + \frac{1}{L}f(h_{k,i}^L(t), Z_{k+1}^L(t))\| \leqslant (1 + \frac{K_1}{L})\|h_{k,i}^L(t)\|,$$

since $f(\cdot, Z_{k+1}^L(t))$ is $K_1$-Lipschitz, where $K_1$ is defined by (18), and $f(0, Z_{k+1}^L(t)) = 0$. Therefore, for any $k \in \{1, \dots, L\}$,

$$\|h_{k,i}^L(t)\| \leqslant e^{K_1}\|h_{0,i}^L(t)\| = e^{K_1}\|A^L(t)x_i\| \leqslant e^{K_1}M M_X. \tag{21}$$

This bound shows that the pair $(h_{k,i}^L(t), Z_{k+1}^L(t))$ belongs to the compact $E$ defined in (19) for every $t \in [0, T]$, $k \in \{1, \dots, L\}$, and $i \in \{1, \dots, n\}$. In particular, $\|\partial_2 f(h_{k-1,i}^L(t), Z_k^L(t))\|_2 \leqslant K$, and

$$\left\|\partial_2 f(h_{k,i}^L(t), Z_{k+1}^L(t)) - \partial_2 f(h_{k-1,i}^L(t), Z_k^L(t))\right\|_2$$
$$\leqslant K'\|h_{k,i}^L(t) - h_{k-1,i}^L(t)\| + K'\|Z_{k+1}^L(t) - Z_k^L(t)\|.$$

Returning to (20), we obtain

$$\left\|\frac{dZ_{k+1}^L}{dt}(t) - \frac{dZ_k^L}{dt}(t)\right\| \leqslant \frac{1}{n}\sum_{i=1}^n K\|p_{k,i}^L(t) - p_{k-1,i}^L(t)\|$$
$$+ K'\|p_{k-1,i}^L(t)\|\big(\|h_{k,i}^L(t) - h_{k-1,i}^L(t)\| + \|Z_{k+1}^L(t) - Z_k^L(t)\|\big).$$

For $k \in \{1, \ldots, L\}$ and $i \in \{1, \ldots, n\}$,

$$\left\|p_{k,i}^L(t) - p_{k-1,i}^L(t)\right\| = \frac{1}{L}\big\|\partial_1 f(h_{k-1,i}^L(t), Z_k^L(t))p_{k,i}^L(t)\big\| \leqslant \frac{K}{L}\left\|p_{k,i}^L(t)\right\|,$$

and, similarly,

$$\left\|h_{k,i}^L(t) - h_{k-1,i}^L(t)\right\| = \frac{1}{L}\|f(h_{k-1,i}^L(t), Z_k^L(t))\| \leqslant \frac{K}{L}\left\|h_{k-1,i}^L(t)\right\| \leqslant \frac{Ke^K M M_X}{L}.$$

Thus,

$$\left\|\frac{dZ_{k+1}^L}{dt}(t) - \frac{dZ_k^L}{dt}(t)\right\| \leqslant \frac{1}{n}\sum_{i=1}^n \|p_{k,i}^L(t)\|\Big(\frac{K^2}{L} + \frac{K'K}{L}e^K M M_X + K'\|Z_{k+1}^L(t) - Z_k^L(t)\|\Big).$$

Moreover, for $k \in \{0, \ldots, L\}$ and $i \in \{1, \ldots, n\}$,

$$\|p_{k,i}^L(t)\| \leqslant \|p_{k+1,i}^L(t)\| + \frac{1}{L}\big\|\partial_1 f(h_{k,i}^L(t), Z_{k+1}^L(t))p_{k+1,i}^L(t)\big\| \leqslant \|p_{k+1,i}^L(t)\| + \frac{K}{L}\|p_{k+1,i}^L(t)\|.$$

Hence

$$\|p_{k,i}^L(t)\| \leqslant e^K\|p_{L,i}^L(t)\| = 2e^K\|B^L(t)^\top(F^L(x_i; t) - y_i)\|$$
$$\leqslant 2e^K M\big(\|B^L(t)h_{L,i}^L(t)\| + \|y_i\|\big) \leqslant 2e^K M(e^K M^2 M_X + M_Y),$$

where we use (17) and (21) for the last inequality. Putting all the pieces together, we obtain

$$\left\|\frac{dZ_k^L}{dt}(t) - \frac{dZ_{k+1}^L}{dt}(t)\right\| \leqslant \alpha\|Z_k^L(t) - Z_{k+1}^L(t)\| + \frac{\beta}{L}.$$

Integrating between $0$ and $t$, we see that

$$\|Z_{k+1}^L(t) - Z_k^L(t)\| \leqslant \|Z_{k+1}^L(0) - Z_k^L(0)\| + \frac{\beta t}{L} + \int_0^t \alpha\|Z_k^L(\tau) - Z_{k+1}^L(\tau)\|d\tau.$$

Applying Grönwall's inequality (see, e.g., Dragomir, 2003), we conclude that $\|Z_{k+1}^L(t) - Z_k^L(t)\| \leqslant (\|Z_{k+1}^L(0) - Z_k^L(0)\| + \frac{\beta T}{L})e^{\alpha T}$, as desired. □

**Remark 1.** *Clipping is used in our approach to constraint the gradients to live in a ball. It is merely a technical assumption to avoid blow-up of the weights during training. However, in any scenario where we know that the weights do not blow up, clipping is not required. A first example of such a scenario is under the Polyak-Łojasiewicz condition (see below). Another scenario is by using gradient flow with momentum instead of vanilla gradient flow. This is a setup closer to Adam (Kingma & Ba, 2015), which is a very used optimizer in practice. One might then show a similar result to Theorem 4, without clipping, because the gradient updates in the momentum case are bounded by construction.*

## A.2  THE TRAINED WEIGHTS ARE BOUNDED UNDER THE LOCAL PL CONDITION

**Proposition 8.** *Consider the residual network (12) initialized as explained in Appendix A and trained with the gradient flow (4) on $[0, \infty]$. Then, for $M > 0$, there exists $\mu > 0$ such that, if the residual network satisfies the $(M, \mu)$-local PL condition (1) around its initialization for any $L \in \mathbb{N}^*$, then:*

*(i) The gradient flow is well defined on $\mathbb{R}_+$, and, for $t \in \mathbb{R}_+$, $L \in \mathbb{N}^*$, and $k \in \{1, \ldots, L\}$,*

$$\|A^L(t)\|_F \leqslant M_A, \quad \|Z_k^L(t)\| \leqslant M_Z, \quad and \quad \|B^L(t)\|_F \leqslant M_B,$$

*where*

$$M_A = \|A^{\text{init}}\|_2 + M, \quad M_Z = \sup_{s \in [0,1]} \|Z^{\text{init}}(s)\| + M, \quad and \quad M_B = \|B^{\text{init}}\|_2 + M.$$

(ii) *There exists $\tilde{K} > 0$ such that, for $t \in \mathbb{R}_+$, $L \in \mathbb{N}^*$, and $k \in \{1, \dots, L\}$,*

$$\|Z_k^L(t) - Z_{k+1}^L(t)\| \leqslant \frac{\tilde{K}}{L}.$$

(iii) *There exists a bounded integrable function $b : \mathbb{R}_+ \to \mathbb{R}$ such that, for $t \in \mathbb{R}_+$, $L \in \mathbb{N}^*$, and $k \in \{1, \dots, L\}$,*

$$\max\left(\left\|\frac{dA^L}{dt}(t)\right\|, \left\|\frac{dZ_k^L}{dt}(t)\right\|, \left\|\frac{dB^L}{dt}(t)\right\|\right) \leqslant b(t)$$

(iv) *$A^L(t)$, $B^L(t)$, and $Z_k^L(t)$ admit a limit uniformly over $L \in \mathbb{N}^*$ and $k \in \{1, \dots, L\}$ as $t \to \infty$.*

(v) *For $t \in \mathbb{R}_+$ and $L \in \mathbb{N}^*$, $\ell^L(t) \leqslant e^{-\mu t}\ell^L(0)$.*

*Moreover, the following expression for $\mu$ hold:*

$$\mu = \max(M_B K, M_B M_X, M_A M_X)\frac{8e^K}{M} \sup_{L \in \mathbb{N}^*} \sqrt{\ell^L(0)}, \tag{22}$$

*where*

$$M_X = \sup_{x \in \mathcal{X}} \|x\|, \quad K_1 = \sup_{\|z\| \leqslant M_Z} \left\|\partial_1 f(h, z)\right\|$$

$$E = \{(h, z) \in \mathbb{R}^d \times \mathbb{R}^p, \|h\| \leqslant e^{K_1} M_A M_X, \|z\| \leqslant M_Z\}$$

$$K_2 = \sup_{(h,z) \in E} \left\|\partial_2 f(h, z)\right\|, \quad K = \max(K_1, K_2).$$

*Proof.* Let $M > 0$, $\mu$ defined by (22), and assume that the residual network satisfies the $(M, \mu)$-local PL condition (1) around its initialization for any $L \in \mathbb{N}^*$.

The time-independent dynamics

$$(A^L, Z_k^L, B^L) \mapsto \left(-\frac{\partial \ell^L}{\partial A^L}, -L\frac{\partial \ell^L}{\partial Z_k^L}, -\frac{\partial \ell^L}{\partial B^L}\right)$$

defining the gradient flow (5) are locally Lipschitz continuous, hence the gradient flow is defined on a maximal interval $[0, T_{\max})$ by the Picard-Lindelöf theorem (see Lemma 16). Let us show by contradiction that $T_{\max} = \infty$. Assume that $T_{\max} < \infty$. If this is true, again by the Picard-Lindelöf theorem, we know that the parameters diverge to infinity at $T_{\max}$. In particular, there exist $t \in (0, T_{\max})$ and $k \in \{1, \dots, L\}$ such that

$$\|A^L(t) - A^L(0)\|_F > M \text{ or } \|Z_k^L(t) - Z_k^L(0)\| > M \text{ or } \|B^L(t) - B^L(0)\|_F > M.$$

Let $t^* \in (0, T_{\max})$ be the infimum of such times $t$. Then, for $t < t^*$ and $k \in \{1, \dots, L\}$,

$$\|A^L(t) - A^L(0)\|_F \leqslant M \text{ and } \|Z_k^L(t) - Z_k^L(0)\| \leqslant M \text{ and } \|B^L(t) - B^L(0)\|_F \leqslant M, \tag{23}$$

and, by continuity of $A^L$, $B^L$, and $Z_k^L$, these inequalities also hold for $t = t^*$. By definition, this means that the $(M, \mu)$-local PL condition is satisfied for $t \leqslant t^*$, and ensures that

$$\left\|\frac{\partial \ell^L}{\partial A^L}(t)\right\|_F^2 + L\sum_{k=1}^{L}\left\|\frac{\partial \ell^L}{\partial Z_k^L}(t)\right\|^2 + \left\|\frac{\partial \ell^L}{\partial B^L}(t)\right\|_F^2 \geqslant \mu\ell^L(t).$$

Therefore, by definition of the gradient flow (4),

$$\frac{d\ell^L}{dt}(t) = \left\langle\frac{\partial \ell^L}{\partial A^L}(t), \frac{dA^L}{dt}(t)\right\rangle + \sum_{k=1}^{L}\left\langle\frac{\partial \ell^L}{\partial Z_k^L}(t), \frac{dZ_k^L}{dt}(t)\right\rangle + \left\langle\frac{\partial \ell^L}{\partial B^L}(t), \frac{dB^L}{dt}(t)\right\rangle$$

$$= -\left\|\frac{\partial \ell^L}{\partial A^L}(t)\right\|_F^2 - L\sum_{k=1}^{L}\left\|\frac{\partial \ell^L}{\partial Z_k^L}(t)\right\|^2 - \left\|\frac{\partial \ell^L}{\partial B^L}(t)\right\|_F^2$$

$$\leqslant -\mu\ell^L(t).$$

Thus, by Grönwall's inequality, for $t \leqslant t^*$,

$$\ell^L(t) \leqslant e^{-\mu t} \ell^L(0). \tag{24}$$

Furthermore, by (23) and the definition of $M_A$, $M_B$, $M_Z$, we have, for $t \leqslant t^*$ and $k \in \{1, \ldots, L\}$,

$$\|A^L(t)\|_F \leqslant M_A, \quad \|Z_k^L(t)\| \leqslant M_Z, \quad \text{and} \quad \|B^L(t)\|_F \leqslant M_B.$$

A quick scan through the proof of Proposition 7 reveals that by similar arguments, we have, for $t \leqslant t^*$, $k \in \{1, \ldots, L\}$, and $i \in \{1, \ldots, n\}$,

$$(h_{k-1,i}^L(t), Z_k^L(t)) \in E \quad \text{and} \quad \|p_{k-1,i}^L(t)\| \leqslant 2e^K \|p_{L,i}^L(t)\| \leqslant 2e^K M_B \|F^L(x_i; t) - y_i\|.$$

Thus, for $k \in \{0, \ldots, L\}$,

$$\frac{1}{n} \sum_{i=1}^n \|p_{k,i}^L(t)\| \leqslant \frac{2e^K M_B}{n} \sum_{i=1}^n \|F^L(x_i; t) - y_i\| \leqslant 2e^K M_B \sqrt{\ell^L(t)} \leqslant 2e^K M_B e^{-\frac{\mu t}{2}} \sqrt{\ell^L(0)}, \tag{25}$$

where the second inequality is a consequence of the Cauchy-Schwartz inequality. Let us now bound $\|Z_k^L(t^*) - Z_k^L(0)\|$. We have, for $k \in \{1, \ldots, L\}$,

$$\begin{aligned}
\|Z_k^L(t^*) - Z_k^L(0)\| &\leqslant \int_0^{t^*} \left\| \frac{dZ_k^L}{dt}(t) \right\| dt \\
&\leqslant \frac{1}{n} \sum_{i=1}^n \int_0^{t^*} \left\| \partial_2 f(h_{k-1,i}^L(t), Z_k^L(t))^\top p_{k,i}^L(t) \right\| dt \\
&\qquad \text{(by (15)).} \\
&\leqslant \frac{K}{n} \sum_{i=1}^n \int_0^{t^*} \|p_{k,i}^L(t)\| dt,
\end{aligned}$$

since $(h_{k-1,i}^L(t), Z_k^L(t)) \in E$ and $\|\partial_2 f(h, z)\| \leqslant K$ for $(h, z) \in E$. Therefore, by (25),

$$\|Z_k^L(t^*) - Z_k^L(0)\| \leqslant 2Ke^K M_B \int_0^{t^*} e^{-\frac{\mu t}{2}} \sqrt{\ell^L(0)} dt \leqslant \frac{4Ke^K M_B}{\mu} \sqrt{\ell^L(0)} \leqslant \frac{M}{2},$$

where the last inequality is a consequence of the definition of $\mu$. Similarly, by (14) and (25),

$$\begin{aligned}
\|A^L(t^*) - A^L(0)\|_F &\leqslant \int_0^{t^*} \left\| \frac{dA^L}{dt}(t) \right\|_F dt \\
&\leqslant \int_0^{t^*} \frac{1}{n} \sum_{i=1}^n \left\| p_{0,i}^L(t) x_i^\top \right\|_F dt \\
&\leqslant 2e^K M_B M_X \sqrt{\ell^L(0)} \int_0^{t^*} e^{-\frac{\mu t}{2}} dt \\
&\leqslant \frac{4e^K M_B M_X}{\mu} \sqrt{\ell^L(0)} \\
&\leqslant \frac{M}{2}.
\end{aligned}$$

Finally, by (16),

$$\begin{aligned}
\|B^L(t^*) - B(0)\|_F &\leqslant \int_0^{t^*} \left\| \frac{dB^L}{dt}(t) \right\|_F dt \\
&\leqslant \int_0^{t^*} \frac{2}{n} \sum_{i=1}^n \|(F^L(x_i; t) - y_i) h_{L,i}^L(t)^\top\|_F dt \\
&\leqslant 2e^K M_A M_X \sqrt{\ell^L(0)} \int_0^{t^*} e^{-\frac{\mu t}{2}} dt \\
&\leqslant \frac{4e^K M_A M_X}{\mu} \sqrt{\ell^L(0)} \\
&\leqslant \frac{M}{2},
\end{aligned}$$

where the third inequality is a consequence of the Cauchy-Schwartz inequality and of the fact that $\|h_{L,i}^L(t)\| \leqslant e^K M_A M_X$. By continuity of $A^L$, $Z_k^L$, and $B^L$, these three bounds contradict the definition of $t^*$. We conclude that $T_{\max} = \infty$ and that the parameters stay within a ball of radius $M$ of their initialization, yielding the inequalities, for $t \in \mathbb{R}_+$, $L \in \mathbb{N}^*$, and $k \in \{1, \ldots, L\}$,

$$\|A^L(t)\|_F \leqslant M_A, \quad \|B^L(t)\|_F \leqslant M_B, \quad \|Z_k^L(t)\| \leqslant M_Z.$$

This proves statement $(i)$ of the proposition. Moreover, the analysis above show that the derivatives of $A^L$, $Z_k^L$, and $B^L$ are bounded by a bounded integrable function independent of $L$ and $k$. This shows $(iii)$, together with the fact that the functions $A^L(t)$, $Z_k^L(t)$, and $B^L(t)$ admit limits as $t \to \infty$. Furthermore, the convergence towards their limit is uniform over $L$ and $k$, as we show for example for $A^L(t)$. If we denote by $A_\infty^L$ its limit, and apply the same steps as for bounding $\|A^L(t^*) - A^L(0)\|_F$, we obtain, for any $t \geqslant 0$,

$$\begin{aligned}
\|A_\infty^L - A^L(t)\|_F &\leqslant \int_t^\infty \left\| \frac{dA^L}{d\tau}(\tau) \right\|_F d\tau \\
&\leqslant 2e^K M_B M_X \sqrt{\ell^L(0)} \int_t^\infty e^{\frac{-\mu\tau}{2}} d\tau \\
&= \frac{4e^K M_B M_X}{\mu} e^{\frac{-\mu t}{2}} \sqrt{\ell^L(0)} \\
&\leqslant \frac{M}{2} e^{\frac{-\mu t}{2}},
\end{aligned}$$

where the last inequality comes from the definition of $\mu$. The bound is independent of $L$, proving statement $(iv)$. Statement $(v)$ readily follows from (24).

To complete the proof, it remains to prove statement $(ii)$ by bounding the differences $\|Z_{k+1}^L(t) - Z_k^L(t)\|$. Now that we know that the weights are bounded, we can follow the same steps as in the proof of Proposition 7 and show the existence of $C_1, C_2 > 0$ such that

$$\left\| \frac{dZ_{k+1}^L}{dt}(t) - \frac{dZ_k^L}{dt}(t) \right\| \leqslant \frac{1}{n} \sum_{i=1}^n \|p_{k,i}^L(t)\| \left( \frac{C_1}{L} + C_2 \|Z_{k+1}^L(t) - Z_k^L(t)\| \right).$$

Using (25), we obtain

$$\left\| \frac{dZ_{k+1}^L}{dt}(t) - \frac{dZ_k^L}{dt}(t) \right\| \leqslant 2e^K M_B e^{-\frac{\mu t}{2}} \sqrt{\ell^L(0)} \left( \frac{C_1}{L} + C_2 \|Z_{k+1}^L(t) - Z_k^L(t)\| \right).$$

Integrating between $0$ and $t$, we obtain

$$\begin{aligned}
\|Z_{k+1}^L(t) - Z_k^L(t)\| &\leqslant \|Z_{k+1}^L(0) - Z_k^L(0)\| + \int_0^t 2e^K M_B e^{-\frac{\mu\tau}{2}} \sqrt{\ell^L(0)} \frac{C_1}{L} d\tau \\
&\quad + \int_0^t 2e^K M_B e^{-\frac{\mu\tau}{2}} \sqrt{\ell^L(0)} C_2 \|Z_{k+1}^L(\tau) - Z_k^L(\tau)\| d\tau \\
&\leqslant \|Z_{k+1}^L(0) - Z_k^L(0)\| + \frac{C_1 M}{2M_X L} \\
&\quad + \int_0^t 2e^K M_B e^{-\frac{\mu\tau}{2}} \sqrt{\ell^L(0)} C_2 \|Z_{k+1}^L(\tau) - Z_k^L(\tau)\| d\tau,
\end{aligned}$$

where the second inequality uses the definition of $\mu$. By Grönwall's inequality,

$$\begin{aligned}
\|Z_{k+1}^L(t) - Z_k^L(t)\| &\leqslant \left( \|Z_{k+1}^L(0) - Z_k^L(0)\| + \frac{C_1 M}{2M_X L} \right) \exp\left( \int_0^t 2e^K M_B e^{-\frac{\mu\tau}{2}} \sqrt{\ell^L(0)} C_2 d\tau \right) \\
&\leqslant \left( \|Z_{k+1}^L(0) - Z_k^L(0)\| + \frac{C_1 M}{2M_X L} \right) \exp\left( \frac{C_2 M}{2M_X} \right),
\end{aligned}$$

again by definition of $\mu$. Finally, since $Z_k^L(0) = Z^{\text{init}}(\frac{k}{L})$ and $Z^{\text{init}}$ is Lipschitz continuous, this proves the existence of $\tilde{K} > 0$ (independent of $L$, $t$ and $k$) such that $\|Z_{k+1}^L(t) - Z_k^L(t)\| \leqslant \frac{\tilde{K}}{L}$, which yields statement $(ii)$. $\qquad\square$

### A.3 GENERALIZED ARZELÀ–ASCOLI THEOREM

**Proposition 9** (Generalized Arzelà–Ascoli theorem). *Let $I \subseteq \mathbb{R}_+$ be an interval, and $(Z_k^L)_{L \in \mathbb{N}^*, 1 \leqslant k \leqslant L}$ be a family of $\mathcal{C}^1$ functions from $I$ to $\mathbb{R}^p$. Define*

$$\mathcal{Z}^L : [0,1] \times I \to \mathbb{R}^p, \ (s,t) \mapsto \mathcal{Z}^L(s,t) = Z_{\lfloor (L-1)s \rfloor + 1}^L(t).$$

*Assume that there exist a constant $C > 0$ and a bounded integrable function $b : I \to \mathbb{R}$ such that the following statements hold for any $t \in I$ and $L \in \mathbb{N}^*$:*

*(i) For $k \in \{1, \ldots, L-1\}$, $\|Z_{k+1}^L(t) - Z_k^L(t)\| \leqslant \frac{C}{L}$,*

*(ii) For $k \in \{1, \ldots, L\}$, $\|Z_k^L(t)\| \leqslant C$ and $\|\frac{dZ_k^L}{dt}(t)\| \leqslant b(t)$.*

*Then there exist a subsequence $(\mathcal{Z}^{\phi(L)})_{L \in \mathbb{N}^*}$ of $(\mathcal{Z}^L)_{L \in \mathbb{N}^*}$ and a Lipschitz continuous function $\mathcal{Z}^\phi : [0,1] \times I \to \mathbb{R}^p$ such that $\mathcal{Z}^{\phi(L)}(s,t)$ tends to $\mathcal{Z}^\phi(s,t)$ uniformly over $s$ and $t$.*

Note that if $I$ is a compact interval, then the existence of a (uniformly) convergent subsequence is guaranteed by the standard Arzelà–Ascoli theorem. Indeed, the uniform equicontinuity is a consequence of assumptions $(i)$ and $(ii)$, while $(ii)$ provides a uniform bound. However, if $I$ is not compact, more involved arguments are needed.

*Proof.* Assume, without loss of generality, that $b$ is also bounded by $C$. According to assumption $(i)$, for $t \in I$ and $i, j \in \{1, \ldots, L\}$,

$$\|Z_i^L(t) - Z_j^L(t)\| \leqslant \frac{C|i-j|}{L}.$$

Also, according to $(ii)$, for $t, t' \in I$ and $k \in \{1, \ldots, L\}$,

$$\|Z_k^L(t) - Z_k^L(t')\| = \left\| \int_{t'}^t \frac{dZ_k^L}{d\tau}(\tau)d\tau \right\| \leqslant C|t - t'|.$$

It follows that, for $s, s' \in [0,1]$ and $t, t' \in I$,

$$\|\mathcal{Z}^L(s,t) - \mathcal{Z}^L(s',t')\| \leqslant \|\mathcal{Z}^L(s,t) - \mathcal{Z}^L(s,t')\| + \|\mathcal{Z}^L(s,t') - \mathcal{Z}^L(s',t')\|$$

$$\leqslant C|t - t'| + \frac{C|\lfloor (L-1)s \rfloor - \lfloor (L-1)s' \rfloor|}{L}.$$

Therefore, with some simple algebra, we obtain

$$\|\mathcal{Z}^L(s,t) - \mathcal{Z}^L(s',t')\| \leqslant C|t - t'| + C|s - s'| + \frac{C}{L}. \tag{26}$$

The statement of the proposition is then a consequence of the next three steps.

**There exists a convergent subsequence of $(\mathcal{Z}^L(s,t))_{L \in \mathbb{N}^*}$.** First, let $((s_i, t_i))_{i \in \mathbb{N}} = (\mathbb{Q} \cap [0,1]) \times (\mathbb{Q} \cap I)$. By $(ii)$, the sequence $(\mathcal{Z}^L(s_i, t_i))_{L \in \mathbb{N}^*, i \in \mathbb{N}}$ is bounded. It is therefore possible to construct by a diagonal procedure a subsequence $(\mathcal{Z}^{\phi(L)})_{L \in \mathbb{N}^*}$ such that, for each $i \in \mathbb{N}$, $(\mathcal{Z}^{\phi(L)}(s_i, t_i))_{L \in \mathbb{N}^*}$ is a convergent sequence.

Let us now show that $(\mathcal{Z}^{\phi(L)}(s,t))_{L \in \mathbb{N}^*}$ converges for any $s \in [0,1]$ and $t \in I$, by proving that it is a Cauchy sequence in the complete metric space $\mathbb{R}^p$. Let $\varepsilon > 0$, $s \in [0,1]$, and $t \in I$. Since $((s_i, t_i))_{i \in \mathbb{N}}$ is dense in $[0,1] \times I$, there exists some $j \in \mathbb{N}$ such that $|s_j - s| \leqslant \varepsilon$ and $|t_j - t| \leqslant \varepsilon$. Then, for $L, M \in \mathbb{N}^*$, we have

$$\|\mathcal{Z}^{\phi(L)}(s,t) - \mathcal{Z}^{\phi(M)}(s,t)\|$$

$$\leqslant \|\mathcal{Z}^{\phi(L)}(s,t) - \mathcal{Z}^{\phi(L)}(s_j, t_j)\| + \|\mathcal{Z}^{\phi(L)}(s_j, t_j) - \mathcal{Z}^{\phi(M)}(s_j, t_j)\|$$

$$+ \|\mathcal{Z}^{\phi(M)}(s_j, t_j) - \mathcal{Z}^{\phi(M)}(s,t)\|$$

$$\leqslant 2C\varepsilon + \frac{C}{\phi(L)} + \|\mathcal{Z}^{\phi(L)}(s_j, t_j) - \mathcal{Z}^{\phi(M)}(s_j, t_j)\| + 2C\varepsilon + \frac{C}{\phi(M)},$$

where we used inequality (26) twice. Since $(\mathcal{Z}^{\phi(L)}(s_j, t_j))_{L \in \mathbb{N}^*}$ is a convergent sequence, it is a Cauchy sequence. Thus, the bound can be made arbitrarily small for $L, M$ large enough. This shows that $(\mathcal{Z}^{\phi(L)}(s,t))_{L \in \mathbb{N}^*}$ is also a Cauchy sequence. It is therefore convergent, and we denote by $\mathcal{Z}^\phi(s,t)$ its limit.

**The function $\mathcal{Z}^\phi$ is Lipschitz continuous.** By considering (26) for the subsequence $\phi(L)$ and letting $L \to \infty$, we have that, for any $s, s' \in [0, 1]$ and $t, t' \in I$,

$$\|\mathcal{Z}^\phi(s, t) - \mathcal{Z}^\phi(s', t')\| \leqslant C(|s - s'| + |t - t'|). \tag{27}$$

**The convergence of $(\mathcal{Z}^{\phi(L)}(s, t))_{L \in \mathbb{N}^*}$ to $\mathcal{Z}^\phi(s, t)$ is uniform over $s$ and $t$.** Let $\varepsilon > 0$, $s \in [0, 1]$, and $t \in I$. Then, by (26) and (27), it is possible to find $\delta > 0$ such that, for any $s', s'' \in [0, 1]$ and $t', t'' \in I$ satisfying $|s' - s''| \leqslant \delta$ and $|t' - t''| \leqslant \delta$,

$$\|\mathcal{Z}^{\phi(L)}(s', t') - \mathcal{Z}^{\phi(L)}(s'', t')\| \leqslant \varepsilon + \frac{C}{\phi(L)} \quad \text{and} \quad \|\mathcal{Z}^\phi(s', t') - \mathcal{Z}^\phi(s'', t')\| \leqslant \varepsilon, \tag{28}$$

and

$$\|\mathcal{Z}^{\phi(L)}(s', t') - \mathcal{Z}^{\phi(L)}(s', t'')\| \leqslant \varepsilon + \frac{C}{\phi(L)} \quad \text{and} \quad \|\mathcal{Z}^\phi(s', t') - \mathcal{Z}^\phi(s', t'')\| \leqslant \varepsilon. \tag{29}$$

Furthermore, there exists a finite set $\{s_1, \ldots, s_S\} \subset [0, 1]$ such that

$$[0, 1] \subset \bigcup_{i=1}^{S} (s_i - \delta, s_i + \delta).$$

In the sequel, we denote by $s^*$ an element of $\{s_1, \ldots, s_S\}$ that is at distance at most $\delta$ from $s$.

If $I$ is unbounded, then, by assumption $(ii)$ and since $b$ is integrable, there exists some $t_0 > 0$ such that, for $t \geqslant t_0$,

$$\|\mathcal{Z}^{\phi(L)}(s, t) - \mathcal{Z}^{\phi(L)}(s, t_0)\| \leqslant \int_{t_0}^{t} \left\| \frac{d}{dt} Z_{\lfloor (\phi(L)s - 1) \rfloor + 1}^{\phi(L)}(\tau) \right\| d\tau \leqslant \int_{t_0}^{t} b(\tau) d\tau \leqslant \varepsilon. \tag{30}$$

The same inequality holds for $\mathcal{Z}^\phi$ by letting $L$ tend to infinity. If $I$ is bounded, we simply let $t_0 = \sup I$.

We may then pick a finite set $\{t_1, \ldots, t_T\} \subset [0, t_0]$ such that

$$[0, t_0] \subset \bigcup_{i=1}^{T} (t_i - \delta, t_i + \delta).$$

Two cases may arise depending on the value of $t$. If $t \in [0, t_0]$, then there exists an element of the set $\{t_1, \ldots, t_T\}$ at distance at most $\delta$ from $t$, and we denote it by $t^*$. If $t > t_0$, we let $t^* = t_0$. According to (29) and (30), we then have in both cases that

$$\|\mathcal{Z}^{\phi(L)}(s, t) - \mathcal{Z}^{\phi(L)}(s, t^*)\| \leqslant \varepsilon + \frac{C}{\phi(L)} \quad \text{and} \quad \|\mathcal{Z}^\phi(s, t) - \mathcal{Z}^\phi(s, t^*)\| \leqslant \varepsilon. \tag{31}$$

To conclude, we have to bound the term $\|\mathcal{Z}^{\phi(L)}(s, t) - \mathcal{Z}^\phi(s, t)\|$ uniformly over $s$ and $t$. We first have

$$\begin{aligned}
\|\mathcal{Z}^{\phi(L)}(s, t) &- \mathcal{Z}^\phi(s, t)\| \\
&\leqslant \|\mathcal{Z}^{\phi(L)}(s, t) - \mathcal{Z}^{\phi(L)}(s, t^*)\| + \|\mathcal{Z}^{\phi(L)}(s, t^*) - \mathcal{Z}^\phi(s, t^*)\| \\
&\quad + \|\mathcal{Z}^\phi(s, t^*) - \mathcal{Z}^\phi(s, t)\| \\
&\leqslant 2\varepsilon + \frac{C}{\phi(L)} + \|\mathcal{Z}^{\phi(L)}(s, t^*) - \mathcal{Z}^\phi(s, t^*)\|,
\end{aligned}$$

where the last inequality is a consequence of (31). The last term can be bounded as follows:

$$\begin{aligned}
\|\mathcal{Z}^{\phi(L)}(s, t^*) &- \mathcal{Z}^\phi(s, t^*)\| \\
&\leqslant \|\mathcal{Z}^{\phi(L)}(s, t^*) - \mathcal{Z}^{\phi(L)}(s^*, t^*)\| + \|\mathcal{Z}^{\phi(L)}(s^*, t^*) - \mathcal{Z}^\phi(s^*, t^*)\| \\
&\quad + \|\mathcal{Z}^\phi(s^*, t^*) - \mathcal{Z}^\phi(s, t^*)\| \\
&\leqslant 2\varepsilon + \frac{C}{\phi(L)} + \max_{i \in \{1, \ldots, S\}} \|\mathcal{Z}^{\phi(L)}(s_i, t^*) - \mathcal{Z}^\phi(s_i, t^*)\|,
\end{aligned}$$

by using (28) and the fact that $s^* \in \{s_1, \ldots, s_S\}$. Putting all the pieces together, we finally obtain

$$\|\mathcal{Z}^{\phi(L)}(s,t) - \mathcal{Z}^{\phi}(s,t)\| \leqslant 4\varepsilon + \frac{2C}{\phi(L)} + \max_{i \in \{1, \ldots, S\}, j \in \{1, \ldots, T\}} \|\mathcal{Z}^{\phi(L)}(s_i, t_j) - \mathcal{Z}^{\phi}(s_i, t_j)\|.$$

By taking $L$ large enough, independent of $s$ and $t$, the sum of the last two terms can be made less than $\varepsilon$. Since $\varepsilon$ is arbitrary, this concludes the proof. $\qquad \square$

A consequence of this result is a simplified version for sequences of functions only indexed by $L$ and not $k$, as follows.

**Corollary 10.** *Let $I \subseteq \mathbb{R}_+$ be an interval, and $(Z^L)_{L \in \mathbb{N}^*}$ be a family of $\mathcal{C}^1$ functions from $I$ to $\mathbb{R}^p$. Assume that there exist a constant $C > 0$ and a bounded integrable function $b : I \to \mathbb{R}$ such that, for any $t \in I$ and $L \in \mathbb{N}^*$, $\|Z^L(t)\| \leqslant C$ and $\|\frac{dZ^L}{dt}(t)\| \leqslant b(t)$. Then there exist a subsequence $(Z^{\phi(L)})_{L \in \mathbb{N}^*}$ of $(Z^L)_{L \in \mathbb{N}^*}$ and a function $Z^{\phi} : I \to \mathbb{R}^p$ such that $Z^{\phi(L)}(t)$ tends to $Z^{\phi}(t)$ uniformly over $t$.*

### A.4 CONSISTENCY OF THE EULER SCHEME FOR PARAMETERIZED ODES

**Proposition 11** (Consistency of the Euler scheme for parameterized ODEs.)**.** *Let $(\theta_k^L)_{L \in \mathbb{N}^*, 1 \leqslant k \leqslant L}$ be a bounded family of vectors of $\mathbb{R}^p$, and let*

$$\Theta^L : [0,1] \to \mathbb{R}^p, \ s \mapsto \theta_{\lfloor (L-1)s \rfloor + 1}^L.$$

*Assume that there exists $\Theta : [0,1] \to \mathbb{R}^p$ a Lipschitz continuous function such that $\Theta^L(s)$ tends to $\Theta(s)$ uniformly over $s$. Let $(a^L)_{L \in \mathbb{N}^*}$ be a sequence of vectors in some compact $E \subset \mathbb{R}^d$ converging to $a \in E$. Let $g : \mathbb{R}^d \times \mathbb{R}^p \to \mathbb{R}^d$ be a $\mathcal{C}^1$ function such that $g(0, \cdot) \equiv 0$ and $g(\cdot, \theta)$ is uniformly Lipschitz continuous for $\theta$ in any compact of $\mathbb{R}^p$. Consider the discrete scheme*

$$
\begin{aligned}
u_0^L &= a^L \\
u_{k+1}^L &= u_k^L + \frac{1}{L} g(u_k^L, \theta_{k+1}^L), \quad k \in \{0, \ldots, L-1\}.
\end{aligned}
\tag{32}
$$

*Then $u_{\lfloor Ls \rfloor}^L$ tends to $U(s)$ uniformly over $s \in [0,1]$, where $U$ is the unique solution of the ODE*

$$
\begin{aligned}
U(0) &= a \\
\frac{dU}{ds}(s) &= g(U(s), \Theta(s)), \quad s \in [0,1].
\end{aligned}
\tag{33}
$$

*Moreover, the convergence only depends on the sequence $(a^L)_{L \in \mathbb{N}^*}$ and on its limit $a \in E$ through $(\|a^L - a\|)_{L \in \mathbb{N}^*}$.*

*Proof.* Let $M$ be a bound of the sequence $(\theta_k^L)_{L \in \mathbb{N}^*, 1 \leqslant k \leqslant L}$. By definition of $\Theta^L$, the sequence $(\Theta^L)_{L \in \mathbb{N}^*}$ is also uniformly bounded by $M$, and the same is true for $\Theta$. Then the function $g(\cdot, \Theta(s))$ is uniformly Lipschitz for $s \in [0,1]$. Furthermore, $(U,s) \mapsto g(U, \Theta(s))$ is continuous in $s$ because $g$ and $\Theta$ are continuous. Thus the ODE (33) has a unique solution on $[0,1]$ by the Picard-Lindelöf theorem (see Lemma 16).

Denote by $C$ the uniform Lipschitz constant of $g(\cdot, \theta)$ for $\|\theta\| \leqslant M$. Since $g(0, \cdot) \equiv 0$ and $g(\cdot, \Theta(s))$ is $C$-Lipschitz, one has

$$\left\| \frac{dU}{ds}(s) \right\| = \|g(U(s), \Theta(s))\| \leqslant C\|U(s)\|.$$

Therefore, by Grönwall's inequality,

$$\|U(s)\| \leqslant \|U(0)\| \exp(C) = \|a\| \exp(C) \leqslant D_E \exp(C),$$

where $D_E = \sup_{x \in E} \|x\| < \infty$. A similar reasoning applies to the discrete scheme (32), using the discrete version of Grönwall's inequality. More precisely, for any $k \in \{0, \ldots, L-1\}$,

$$\|u_{k+1}^L\| \leqslant \|u_k^L\| + \frac{1}{L}\|g(u_k^L, \theta_{k+1}^L)\| \leqslant \left(1 + \frac{C}{L}\right)\|u_k^L\|.$$

Thus,
$$\|u_k^L\| \leqslant \|u_0^L\| \exp(C) = \|a^L\| \exp(C) \leqslant D_E \exp(C).$$

Overall, we can consider a restriction of $g$ to a compact set depending only on $M$, $C$, and $E$, which we will still denote by $g$ with a slight abuse of notation. Since $g$ is $\mathcal{C}^1$, it is therefore bounded and Lipschitz continuous, and we still let $C$ be its Lipschitz constant.

For $L \in \mathbb{N}^*$ and $k \in \{0, \dots, L\}$, we denote by $\Delta_k^L$ the gap between the continuous and the discrete schemes, i.e.,
$$\Delta_k^L = \left\| U\left(\frac{k}{L}\right) - u_k^L \right\|.$$

The next step is to recursively bound the size of this gap, first observing that $\Delta_0^L = \|a^L - a\|$. We have that
$$s \mapsto \frac{dU}{ds}(s) = g(U(s), \Theta(s)) \tag{34}$$

is a Lipschitz continuous function with some Lipschitz constant $\tilde{C}$. To see this, just note that $U$ itself is Lipschitz continuous in $s$, since $g$ is bounded, and therefore the function (34) is a composition of Lipschitz continuous functions. In particular, $\frac{dU}{ds}$ is almost everywhere differentiable, and its derivative $\frac{d^2U}{ds^2}(s)$ is bounded in the supremum norm by $\tilde{C}$. As a consequence, for $k \in \{0, \dots, L-1\}$, the Taylor expansion of $U$ on $[\frac{k}{L}, \frac{k+1}{L}]$ takes the form
$$U\left(\frac{k+1}{L}\right) = U\left(\frac{k}{L}\right) + \frac{1}{L}\frac{dU}{ds}\left(\frac{k}{L}\right) + \int_{k/L}^{(k+1)/L} \left(\frac{k+1}{L} - s\right) \frac{d^2U}{ds^2}(s)ds,$$

where the norm of the remainder term is less than $\tilde{C}/L^2$. Therefore,
$$\begin{aligned}
\Delta_{k+1}^L &= \left\| U\left(\frac{k+1}{L}\right) - u_{k+1}^L \right\| \\
&= \left\| U\left(\frac{k}{L}\right) + \frac{1}{L}g\left(U\left(\frac{k}{L}\right), \Theta\left(\frac{k}{L}\right)\right) + \int_{k/L}^{(k+1)/L}\left(\frac{k+1}{L} - s\right)\frac{d^2U}{ds^2}(s)ds \right. \\
&\quad \left. - u_k^L - \frac{1}{L}g(u_k^L, \theta_{k+1}^L) \right\| \\
&\leqslant \left\| U\left(\frac{k}{L}\right) - u_k^L \right\| + \left\| \frac{1}{L}g\left(U\left(\frac{k}{L}\right), \Theta\left(\frac{k}{L}\right)\right) - \frac{1}{L}g(u_k^L, \theta_{k+1}^L) \right\| \\
&\quad + \int_{k/L}^{(k+1)/L}\left(\frac{k+1}{L} - s\right)\left\|\frac{d^2U}{ds^2}(s)\right\|ds \\
&\leqslant \Delta_k^L + \frac{C}{L}\Delta_k^L + \frac{C}{L}\left\|\Theta\left(\frac{k}{L}\right) - \theta_{k+1}^L\right\| + \frac{\tilde{C}}{L^2}.
\end{aligned}$$

In the last inequality, we used the fact that $g$ is $C$-Lipschitz. Since, by definition, $\theta_{k+1}^L = \Theta^L(\frac{k}{L-1})$, we obtain, for $k \in \{0, \dots, L-1\}$,
$$\begin{aligned}
\Delta_{k+1}^L &\leqslant \left(1 + \frac{C}{L}\right)\Delta_k^L + \frac{C}{L}\left\|\Theta\left(\frac{k}{L}\right) - \Theta^L\left(\frac{k}{L-1}\right)\right\| + \frac{\tilde{C}}{L^2} \\
&\leqslant \left(1 + \frac{C}{L}\right)\Delta_k^L + \frac{C}{L}\sup_{s \in [0,1]}\|\Theta(s) - \Theta^L(s)\| + \frac{C}{L}\left\|\Theta\left(\frac{k}{L}\right) - \Theta\left(\frac{k}{L-1}\right)\right\| + \frac{\tilde{C}}{L^2} \\
&\leqslant \left(1 + \frac{C}{L}\right)\Delta_k^L + \frac{C}{L}\sup_{s \in [0,1]}\|\Theta(s) - \Theta^L(s)\| + \frac{CC_\Theta}{L^2} + \frac{\tilde{C}}{L^2},
\end{aligned}$$

where $C_\Theta$ is the Lipschitz constant of $\Theta$. By the discrete Grönwall's inequality, we deduce that, for $k \in \{0, \dots, L-1\}$,
$$\begin{aligned}
\Delta_{k+1}^L &\leqslant \left(\Delta_0^L + \sup_{s \in [0,1]}\|\Theta(s) - \Theta^L(s)\| + \frac{C_\Theta}{L} + \frac{\tilde{C}}{LC}\right)e^C \\
&= \left(\|a^L - a\| + \sup_{s \in [0,1]}\|\Theta(s) - \Theta^L(s)\| + \frac{C_\Theta}{L} + \frac{\tilde{C}}{LC}\right)e^C. \tag{35}
\end{aligned}$$

This shows that the gaps $\Delta_k^L$ converge to zero uniformly over $k \in \{0, \ldots, L\}$ as $L$ tends to infinity.

We conclude by observing that, for any $s \in [0, 1]$,

$$\left\| U(s) - u_{\lfloor Ls \rfloor}^L \right\| \leqslant \left\| U(s) - U\left(\frac{\lfloor Ls \rfloor}{L}\right) \right\| + \left\| U\left(\frac{\lfloor Ls \rfloor}{L}\right) - u_{\lfloor Ls \rfloor}^L \right\| \leqslant \frac{C_U}{L} + \Delta_{\lfloor Ls \rfloor}^L, \qquad (36)$$

where $C_U$ is the Lipschitz constant of $U$. Both terms converge to zero uniformly over $s$ as $L$ tends to infinity. Finally, an inspection of our bounds shows that the convergence only depends on $(a^L)_{L \in \mathbb{N}^*} \in E^{\mathbb{N}^*}$ through $\|a^L - a\|$. $\qquad \square$

The results of Proposition 11 can be extended without much effort to two other related cases. First, the parameters $\theta_k^L$ may depend on some other variable $t$, as long as all assumptions are verified uniformly over $t$. Second, these parameters may converge to some limit parameters as both $L$ and $t$ go to infinity. This is encapsulated in the following two corollaries.

**Corollary 12.** *Let $I \subseteq \mathbb{R}_+$ be an interval. Let $(\theta_k^L)_{L \in \mathbb{N}^*, 1 \leqslant k \leqslant L}$ be a uniformly bounded family of functions from $I$ to $\mathbb{R}^p$, and let*

$$\Theta^L : [0, 1] \times I \to \mathbb{R}^p, \ (s, t) \mapsto \theta_{\lfloor (L-1)s \rfloor + 1}^L(t).$$

*Assume that there exists a function $\Theta : [0, 1] \times I \to \mathbb{R}^p$ such that $\Theta^L(s, t)$ tends to $\Theta(s, t)$ uniformly over $s$ and $t$, and $\Theta(\cdot, t)$ is uniformly Lipschitz continuous for $t \in I$. Let $(a^L)_{L \in \mathbb{N}^*}$ be a family of functions from $I$ to some compact $E \subset \mathbb{R}^d$, uniformly converging to $a : I \to E$. Let $g : \mathbb{R}^d \times \mathbb{R}^p \to \mathbb{R}^d$ be a $\mathcal{C}^1$ function such that $g(0, \cdot) \equiv 0$ and $g(\cdot, \theta)$ is uniformly Lipschitz continuous for $\theta$ in any compact of $\mathbb{R}^p$. Consider the discrete scheme, for $t \in I$,*

$$u_0^L(t) = a^L(t)$$

$$u_{k+1}^L(t) = u_k^L(t) + \frac{1}{L} g(u_k^L(t), \theta_{k+1}^L(t)), \quad k \in \{0, \ldots, L-1\}.$$

*Then $u_{\lfloor Ls \rfloor}^L(t)$ tends to $U(s, t)$ uniformly over $s \in [0, 1]$ and $t \in I$, where $U(\cdot, t)$ is the unique solution of the ODE*

$$U(0, t) = a(t)$$

$$\frac{\partial U}{\partial s}(s, t) = g(U(s, t), \Theta(s, t)), \quad s \in [0, 1].$$

*Moreover, the convergence only depends on the sequence $(a^L)_{L \in \mathbb{N}^*}$ and on its limit $a \in E^I$ through $(\sup_{t \in I} \|a^L(t) - a(t)\|)_{L \in \mathbb{N}^*}$.*

**Corollary 13.** *Let $I \subseteq \mathbb{R}_+$ be an interval. Let $(\theta_k^L)_{L \in \mathbb{N}^*, 1 \leqslant k \leqslant L}$ be a uniformly bounded family of functions from $I$ to $\mathbb{R}^p$, and let*

$$\Theta^L : [0, 1] \times \mathbb{R}_+ \to \mathbb{R}^p, \ (s, t) \mapsto \theta_{\lfloor (L-1)s \rfloor + 1}^L(t).$$

*Assume that there exists a function $\Theta_\infty : [0, 1] \to \mathbb{R}^p$ such that $\Theta^L(s, t)$ tends to $\Theta_\infty(s)$ uniformly over $s$ as $L, t \to \infty$, and $\Theta_\infty$ is Lipschitz continuous. Let $(a^L)_{L \in \mathbb{N}^*}$ be a family of functions from $I$ to some compact $E \subset \mathbb{R}^d$, and converging to $a_\infty \in E$ as $L, t \to \infty$. Let $g : \mathbb{R}^d \times \mathbb{R}^p \to \mathbb{R}^d$ be a $\mathcal{C}^1$ function such that $g(0, \cdot) \equiv 0$ and $g(\cdot, \theta)$ is uniformly Lipschitz continuous for $\theta$ in any compact of $\mathbb{R}^p$. Consider the discrete scheme, for $t \in I$,*

$$u_0^L(t) = a^L(t)$$

$$u_{k+1}^L(t) = u_k^L(t) + \frac{1}{L} g(u_k^L(t), \theta_{k+1}^L(t)), \quad k \in \{0, \ldots, L-1\}.$$

*Then $u_{\lfloor Ls \rfloor}^L(t)$ tends to $U(s)$ uniformly over $s \in [0, 1]$ as $L, t \to \infty$, where $U$ is the unique solution of the ODE*

$$U(0) = a_\infty$$

$$\frac{dU}{ds}(s) = g(U(s), \Theta_\infty(s)), \quad s \in [0, 1].$$

*Moreover, the convergence only depends on the sequence $(a^L)_{L \in \mathbb{N}^*}$ and on its limit $a \in E^I$ through $(\sup_{t \in I} \|a^L(t) - a(t)\|)_{L \in \mathbb{N}^*}$.*

A.5   LARGE-DEPTH CONVERGENCE OF THE GRADIENT FLOW

This section is devoted to proving the main result of Appendix A, namely the large-depth convergence of the gradient flow. The setting we consider encompasses both Section 4.1 (finite training time and clipped gradient flow) and Section 4.2 (arbitrary training time and standard gradient flow). To this end, we consider a training interval $I = [0, T] \subseteq \mathbb{R}_+$, for $T \leqslant \infty$, and the gradient flow formulation (5), which is equivalent to the standard gradient flow (4) if $\pi$ equals the identity. Note that we do not need to assume in the following proof that $\pi$ is bounded (but only Lipschitz continuous). Therefore, the proof also holds in the case where $\pi$ equals the identity.

**Theorem 14.** *Consider the residual network (12) initialized as explained in Appendix A and trained with the gradient flow (5) on $I = [0, T] \subseteq \mathbb{R}_+$, for some $T \in (0, \infty]$. Assume that there exists a unique solution to the gradient flow, such that $(A^L)_{L \in \mathbb{N}^*}$ and $(B^L)_{L \in \mathbb{N}^*}$ each satisfies the assumptions of Corollary 10, and $(Z_k^L)_{L \in \mathbb{N}^*, 1 \leqslant k \leqslant L}$ satisfies the assumptions of Proposition 9. Then the following four statements hold as $L$ **tends to infinity**:*

- (i) *There exist functions $A : I \to \mathbb{R}^{q \times d}$ and $B : I \to \mathbb{R}^{d' \times q}$ such that $A^L(t)$ and $B^L(t)$ converge uniformly over $t \in I$ to $A(t)$ and $B(t)$.*

- (ii) *There exists a Lipschitz continuous function $\mathcal{Z} : [0, 1] \times I \to \mathbb{R}^p$ such that*

$$\mathcal{Z}^L : [0, 1] \times I \to \mathbb{R}^p, \ (s, t) \mapsto \mathcal{Z}^L(s, t) = Z_{\lfloor (L-1)s \rfloor + 1}^L(t)$$

  *converges uniformly over $s \in [0, 1]$ and $t \in I$ to $\mathcal{Z}(s, t)$.*

- (iii) *Uniformly over $s \in [0, 1]$, $t \in I$, and $x \in \mathcal{X}$, the hidden layer $h_{\lfloor Ls \rfloor}^L(t)$ converges to the solution at time $s$ of the neural ODE*

$$H(0, t) = A(t)x$$
$$\frac{\partial H}{\partial s}(s, t) = f(H(s, t), \mathcal{Z}(s, t)), \quad s \in [0, 1].$$

- (iv) *Uniformly over $t \in I$ and $x \in \mathcal{X}$, the output $F^L(x; t)$ converges to $B(t)H(1, t)$.*

*Proof.* According to Proposition 9, there exists a subsequence $(\mathcal{Z}^{\phi(L)})_{L \in \mathbb{N}^*}$ of $(\mathcal{Z}^L)_{L \in \mathbb{N}^*}$ and a Lipschitz continuous function $\mathcal{Z}^\phi : [0, 1] \times I \to \mathbb{R}^p$ such that $\mathcal{Z}^{\phi(L)}(s, t)$ tends to $\mathcal{Z}^\phi(s, t)$ uniformly over $s$ and $t$. Similarly, by Corollary 10, there exists subsequences of $(A^L)_{L \in \mathbb{N}^*}$ and $(B^L)_{L \in \mathbb{N}^*}$ that converge uniformly. With a slight abuse of notation, we still denote these subsequences by $\phi$, and the corresponding limits by $A^\phi$ and $B^\phi$.

In the remainder, we prove the uniqueness of the accumulation point $(Z^\phi, A^\phi, B^\phi)$ by showing that it is the solution of an ODE that satisfies the assumptions of the Picard-Lindelöf theorem. The statements $(i)$ to $(iv)$ then follow easily.

Consider a general input $(x, y) \in \mathcal{X} \times \mathcal{Y}$, and let $H^L(s, t) = h_{\lfloor Ls \rfloor}^L(t)$ (recall that $h_k^L(t)$ is defined by the forward propagation (12)). Corollary 12, with $\theta_k^L = Z_k^{\phi(L)}$, $\Theta = \mathcal{Z}^\phi$, $a^L = A^{\phi(L)}x$, $g = f$, ensures that $H^{\phi(L)}(s, t)$ converges uniformly (over $s$ and $t$) to $H^\phi(s, t)$ that is the solution at time $s$ of the ODE

$$H^\phi(0, t) = A^\phi(t)x$$
$$\frac{\partial H^\phi}{\partial s}(s, t) = f(H^\phi(s, t), \mathcal{Z}^\phi(s, t)), \quad s \in [0, 1].$$

By inspecting the proof of the corollary, we also have that $(h_k^{\phi(L)})_{L \in \mathbb{N}^*, 1 \leqslant k \leqslant \phi(L)}$ and $(H^{\phi(L)})_{L \in \mathbb{N}^*}$ are uniformly bounded and that $H^\phi(\cdot, t)$ is uniformly Lipschitz continuous for $t \in I$.

We now turn our attention to the backpropagation recurrence (13), which defines the backward state $p_k^L(t)$. First observe that the convergence of $H^{\phi(L)}$ implies that

$$p_{\phi(L)}^{\phi(L)}(t) = 2B^{\phi(L)}(t)^\top (B^{\phi(L)}(t)h_{\phi(L)}^{\phi(L)}(t) - y) = 2B^{\phi(L)}(t)^\top (B^{\phi(L)}(t)H^{\phi(L)}(1, t) - y)$$

converges uniformly to $2B^\phi(t)^\top(B^\phi(t)H^\phi(1,t) - y) \in \mathbb{R}^d$. Now, let $P^L(s,t) = p_{\lfloor Ls \rfloor}^L(t)$. We apply again Corollary 12, this time to the backpropagation recurrence (13), with $\theta_k^L = (h_k^{\phi(L)}, Z_k^{\phi(L)})$, $\Theta = (H^\phi, \mathcal{Z}^\phi)$, $g : (p, (h, Z)) \mapsto \partial_1 f(h, Z)p$, and $a^L = 2(B^{\phi(L)})^\top(B^{\phi(L)}H^{\phi(L)}(1,\cdot) - y)$. Let us quickly check that the conditions of the corollary are met:

- The sequence $(h_k^{\phi(L)})_{L \in \mathbb{N}^*, 1 \leqslant k \leqslant \phi(L)}$ is bounded, as noted previously, and the same holds for $(Z_k^{\phi(L)})_{L \in \mathbb{N}^*, 1 \leqslant k \leqslant \phi(L)}$ by the assumptions of Theorem 14.

- The function $H^\phi(\cdot, t)$ is uniformly Lipschitz continuous for $t \in I$, as noted previously, and the same is true for $Z^\phi(\cdot, t)$ since $Z^\phi$ is Lipschitz continuous.

- The function $h_{\lfloor(\phi(L)-1)s\rfloor+1}^{\phi(L)}(t)$ tends to $H^\phi(s,t)$ uniformly over $s$ and $t$, as seen in the beginning of the proof. More precisely, we know that $H^{\phi(L)}(s,t) = h_{\lfloor\phi(L)s\rfloor}^{\phi(L)}(t)$ tends to $H^\phi(s,t)$. Simple algebra and the fact that two successive iterates of (12) are separated by a distance proportional to $1/L$ show that both statements are equivalent. Furthermore, $\mathcal{Z}^{\phi(L)}(s,t)$ tends to $\mathcal{Z}^\phi(s,t)$ uniformly over $s$ and $t$ as noted above.

- The sequence $(a^L)_{L \in \mathbb{N}^*}$ is uniformly bounded, since $B^{\phi(L)}$ and $H^{\phi(L)}(1,\cdot)$ are. It also converges uniformly to $a : t \mapsto 2B^\phi(t)^\top(B^\phi(t)H^\phi(1,t) - y)$.

- The function $g$ is $\mathcal{C}^1$ since $f$ is $\mathcal{C}^2$. We clearly have $g(0,\cdot) \equiv 0$. Finally, $g(\cdot, (h, Z))$ is uniformly Lipschitz continuous for $(h, Z)$ in any compact since $\partial_1 f$ is continuous.

Overall, we obtain that $P^{\phi(L)}(s,t)$ converges uniformly (over $s$ and $t$) to $P^\phi(s,t)$, the solution at time $s$ of the backward ODE

$$P^\phi(1,t) = 2B^\phi(t)^\top(B^\phi(t)H^\phi(1,t) - y)$$
$$\frac{\partial P^\phi}{\partial s}(s,t) = \partial_1 f(H^\phi(s,t), \mathcal{Z}^\phi(s,t))P^\phi(s,t), \quad s \in [0,1].$$

Furthermore, the proof of the corollary shows that $(P^{\phi(L)})_{L \in \mathbb{N}^*}$ is uniformly bounded. Now, recall that the gradient flow for $Z_k^{\phi(L)}(t)$, given by (5) and (15), takes the following form, for $t \in I$ and $k \in \{1, \ldots, \phi(L)\}$,

$$\frac{\partial Z_k^{\phi(L)}(t)}{\partial t} = \pi\Big(-\frac{1}{n}\sum_{i=1}^n \partial_2 f(h_{k-1,i}^{\phi(L)}(t), Z_k^{\phi(L)}(t))^\top p_{k,i}^{\phi(L)}(t)\Big),$$

where the $i$ subscript corresponds to the $i$-th input $x_i$. By definition, for $s \in [0,1]$, $\mathcal{Z}^{\phi(L)}(s,t) = Z_{\lfloor(\phi(L)-1)s\rfloor+1}^{\phi(L)}(t)$. Thus, the equation above can be rewritten, for $s \in [0,1]$ and $t \in I$,

$$\frac{\partial \mathcal{Z}^{\phi(L)}(s,t)}{\partial t} = \pi\Big(-\frac{1}{n}\sum_{i=1}^n \partial_2 f(h_{\lfloor(\phi(L)-1)s\rfloor,i}^{\phi(L)}(t), Z_{\lfloor(\phi(L)-1)s\rfloor+1}^{\phi(L)}(t))^\top p_{\lfloor(\phi(L)-1)s\rfloor+1,i}^{\phi(L)}(t)\Big).$$
(37)

The term inside $\pi$ can be rewritten as

$$-\frac{1}{n}\sum_{i=1}^n \partial_2 f\Big(H_i^{\phi(L)}\Big(\frac{\lfloor(\phi(L)-1)s\rfloor}{\phi(L)}, t\Big), \mathcal{Z}^{\phi(L)}(s,t)\Big)^\top P_i^{\phi(L)}\Big(\frac{\lfloor(\phi(L)-1)s\rfloor+1}{\phi(L)}, t\Big).$$

Since $f$ is $\mathcal{C}^2$, $\partial_2 f$ is locally Lipschitz continuous. Applying the first part of the proof to the specific case of $x_i$, we know that $H_i^{\phi(L)}$ and $P_i^{\phi(L)}$ uniformly bounded, and that $H_i^{\phi(L)}(s,t)$ and $P_i^{\phi(L)}(s,t)$ converge uniformly to $H_i^\phi(s,t)$ and $P_i^\phi(s,t)$. Therefore, the right-hand side of (37) converges uniformly over $s$ and $t$ to

$$\pi\Big(-\frac{1}{n}\sum_{i=1}^n \partial_2 f(H_i^\phi(s,t), \mathcal{Z}^\phi(s,t))^\top P_i^\phi(s,t)\Big).$$

We have just shown the uniform convergence of the derivative in $t$ of $\mathcal{Z}^{\phi(L)}(s,t)$. Furthermore, we know that, for $s \in [0,1]$, the sequence $(t \mapsto \mathcal{Z}^{\phi(L)}(s,t))_{L \in \mathbb{N}^*}$ converges to $\mathcal{Z}^\phi(s,\cdot)$. These two statements imply that $\mathcal{Z}^\phi$ is differentiable with respect to $t$ and that, for $s \in [0,1]$, its derivative satisfies the ordinary differential equation

$$\frac{\partial \mathcal{Z}^\phi(s,t)}{\partial t} = \pi\Big( -\frac{1}{n}\sum_{i=1}^n \partial_2 f(H_i^\phi(s,t), \mathcal{Z}^\phi(s,t))^\top P_i^\phi(s,t) \Big). \tag{38}$$

Moreover, by our initialization scheme,

$$\mathcal{Z}^\phi(s,0) = Z^{\text{init}}(s). \tag{39}$$

A similar approach reveals that $A^\phi(t)$ and $B^\phi(t)$ are differentiable and that they verify the equations

$$\frac{dA^\phi}{dt}(t) = \pi\Big( -\frac{1}{n}\sum_{i=1}^n P_i^\phi(0,t)x_i^\top \Big), \qquad\qquad A^\phi(0) = A^{\text{init}}, \tag{40}$$

$$\frac{dB^\phi}{dt}(t) = \pi\Big( -\frac{2}{n}\sum_{i=1}^n (B^\phi(t)H_i^\phi(1,t) - y_i)H_i^\phi(1,t)^\top \Big), \qquad B^\phi(0) = B^{\text{init}}. \tag{41}$$

The equations (38) to (41) can be seen as an initial value problem whose variables are the function $\mathcal{Z}^\phi(\cdot,t) : [0,1] \to \mathbb{R}^p$ and the matrices $A^\phi(t) \in \mathbb{R}^{q \times d}, B^\phi(t) \in \mathbb{R}^{d' \times q}$. To complete the proof, it remains to show, using the Picard-Lindelöf theorem (see Lemma 16), that there exists a unique solution to this problem. First, note that the space $\mathcal{B}([0,1], \mathbb{R}^p)$ of bounded functions from $[0,1]$ to $\mathbb{R}^p$ endowed with the supremum norm is a Banach space, which is the proper space in which to apply the Picard-Lindelöf theorem. We therefore endow the space of parameters $\mathcal{B}([0,1], \mathbb{R}^p) \times \mathbb{R}^{q \times d} \times \mathbb{R}^{d' \times q}$ with the norm

$$\|(\mathcal{Z}, A, B)\| := \sup_{s \in [0,1]} \|\mathcal{Z}(s)\| + \|A\|_2 + \|B\|_2,$$

which makes it a Banach space. We have to show that the mapping

$$(\mathcal{Z}, A, B) \mapsto \Big( s \mapsto \pi\Big( -\frac{1}{n}\sum_{i=1}^n \partial_2 f(H_i(s), \mathcal{Z}(s))^\top P_i(s) \Big),$$
$$\pi\Big( -\frac{1}{n}\sum_{i=1}^n P_i(0)x_i^\top \Big), \ \pi\Big( -\frac{2}{n}\sum_{i=1}^n (BH_i(1) - y_i)H_i(1)^\top \Big) \Big) \tag{42}$$

is locally Lipschitz continuous with respect to this norm, where we recall that $H_i(s)$ in (42) is the solution at time $s$ of the initial value problem

$$\begin{aligned} H_i(0) &= Ax_i \\ \frac{dH_i}{ds}(s) &= f(H_i(s), \mathcal{Z}(s)), \quad s \in [0,1], \end{aligned} \tag{43}$$

and $P_i(s)$ is the solution at time $s$ of the initial value problem

$$\begin{aligned} P_i(1) &= 2B^\top(BH_i(1) - y_i) \\ \frac{dP_i}{ds}(s) &= \partial_1 f(H_i(s), \mathcal{Z}(s))P_i(s), \quad s \in [0,1]. \end{aligned} \tag{44}$$

To prove that the mapping (42) is locally Lipschitz continuous, we first check that it is well defined. Since $\mathcal{Z}$ is assumed to be only bounded (and not continuous), the solutions of the initial value problems (43) and (44) are well defined in the sense of the Caratheodory conditions, which are given in Lemma 17.

Next, we can show that $(\mathcal{Z}, A, B) \mapsto H_i$ is locally Lipschitz continuous for $i \in \{1, \ldots, n\}$. To do this, consider two sets of parameters $(\mathcal{Z}, A, B)$ and $(\tilde{\mathcal{Z}}, \tilde{A}, \tilde{B})$ belonging to a compact set $D$. Let $H_i$ and $\tilde{H}_i$ denote the corresponding hidden states. As in the proof of Proposition 11, it holds that $H_i$ and $\tilde{H}_i$ belong to some compact set $E$ that depends only on $D$ and $f$. Let $K_f$ be the Lipschitz

constant of the $\mathcal{C}^1$ function $f$ on $E \times D$. Then,

$$\|\tilde{H}_i(s) - H_i(s)\| \leqslant \|\tilde{H}_i(0) - H_i(0)\| + \int_0^s \left\| \frac{d\tilde{H}_i}{dr}(r) - \frac{dH_i}{dr}(r) \right\| dr$$

$$\leqslant \|\tilde{H}_i(0) - H_i(0)\| + \int_0^s \|f(\tilde{H}_i(r), \tilde{\mathcal{Z}}(r)) - f(H_i(r), \mathcal{Z}(r))\| dr.$$

The norm inside the integral can be bounded by

$$\|f(\tilde{H}_i(r), \tilde{\mathcal{Z}}(r)) - f(\tilde{H}_i(r), \mathcal{Z}(r))\| + \|f(\tilde{H}_i(r), \mathcal{Z}(r)) - f(H_i(r), \mathcal{Z}(r))\|$$

$$\leqslant K_f \sup_{r \in [0,1]} \|\tilde{\mathcal{Z}}(r) - \mathcal{Z}(r)\| + K_f \|\tilde{H}_i(r) - H_i(r)\|.$$

Therefore,

$$\|\tilde{H}_i(s) - H_i(s)\| \leqslant \|\tilde{A} - A\|_2 \|x_i\| + K_f \sup_{r \in [0,1]} \|\tilde{\mathcal{Z}}(r) - \mathcal{Z}(r)\| + \int_0^s K_f \|\tilde{H}_i(r) - H_i(r)\| dr.$$

Using Grönwall's inequality, we obtain, for any $s \in [0,1]$,

$$\|\tilde{H}_i(s) - H_i(s)\| \leqslant \left( \|\tilde{A} - A\|_2 \|x_i\| + K_f \sup_{r \in [0,1]} \|\tilde{\mathcal{Z}}(r) - \mathcal{Z}(r)\| \right) \exp(K_f).$$

This shows that the function $(\mathcal{Z}, A, B) \mapsto H_i$ is locally Lipschitz continuous. One proves by similar arguments that the function $(\mathcal{Z}, A, B) \mapsto P_i$ is locally Lipschitz continuous. Thus, overall, the mapping (42) is locally Lipschitz continuous as a composition of locally Lipschitz continuous functions.

The Picard-Lindelöf theorem guarantees the uniqueness of the maximal solution of the initial value problem (38)–(41) in the space $\mathcal{B}([0,1], \mathbb{R}^p) \times \mathbb{R}^{d \times q} \times \mathbb{R}^{d' \times q}$. Since any accumulation point $(\mathcal{Z}^\phi, A^\phi, B^\phi)$ is a solution belonging to this space, this proves the uniqueness of the accumulation point, which we therefore denote as $(\mathcal{Z}, A, B)$.

The uniform convergence of $(\mathcal{Z}^L, A^L, B^L)$ to $(\mathcal{Z}, A, B)$ is then easily shown by contradiction. Suppose that uniform convergence does not hold. If this is true, then there exists a subsequence that stays at distance $\varepsilon > 0$ from $(\mathcal{Z}, A, B)$ (in the sense of the uniform norm). Then arguments similar to the beginning of the proof show the existence of a second accumulation point, which is a contradiction. This shows the uniform convergence, yielding statements $(i)$ and $(ii)$ of the theorem.

Finally, reapplying Corollary 12 with $\theta_k^L = Z_k^L, \Theta = \mathcal{Z}, a^L = A^L x, g = f$, completes the proof by proving statements $(iii)$ and $(iv)$. □

**Training dynamics of the limiting weights.** Interestingly, the proof of Theorem 14 provides us with an explicit description of the evolution of the continuous-depth limiting weights during training. With the notation of the proof, the continuous weights satisfy the training dynamics:

$$\frac{dA}{dt}(t) = \pi\left( -\frac{1}{n} \sum_{i=1}^n P_i(0, t) x_i^\top \right)$$

$$\frac{\partial \mathcal{Z}}{\partial t}(s, t) = \pi\left( -\frac{1}{n} \sum_{i=1}^n \partial_2 f(H_i(s, t), \mathcal{Z}(s, t))^\top P_i(s, t) \right)$$

$$\frac{dB}{dt}(t) = \pi\left( -\frac{2}{n} \sum_{i=1}^n (B(t) H_i(1, t) - y_i) H_i(1, t)^\top \right),$$

where we recall that $H_i(s, t)$ is the solution at time $s$ of the initial value problem

$$H_i(0, t) = A(t) x_i$$

$$\frac{\partial H_i}{\partial s}(s, t) = f(H_i(s, t), \mathcal{Z}(s, t)), \quad s \in [0,1],$$

and $P_i(s, t)$ is the solution at time $s$ of the problem

$$
\begin{aligned}
P_i(1, t) &= 2B(t)^\top (B(t)H_i(1, t) - y_i) \\
\frac{\partial P_i}{\partial s}(s, t) &= \partial_1 f(H_i(s, t), \mathcal{Z}(s, t))P_i(s, t), \quad s \in [0, 1].
\end{aligned}
$$

These equations can be thought of as the continuous-depth equivalent of the backpropagation equations.

### A.6 Existence of the double limit when $L, t$ tend to infinity

**Proposition 15.** *Consider the residual network* (12)*, and assume that:*

(i) *$A^L(t)$, $Z^L_{\lfloor Ls \rfloor}(t)$, and $B^L(t)$ converge uniformly over $L \in \mathbb{N}^*$ and $s \in [0, 1]$ as $t \to \infty$.*

(ii) *$A^L(t)$, $Z^L_{\lfloor Ls \rfloor}(t)$, and $B^L(t)$ converge uniformly over $t \in \mathbb{R}_+$ and $s \in [0, 1]$ as $L \to \infty$.*

(iii) *The loss $\ell^L(t)$ converges to $0$ uniformly over $L \in \mathbb{N}^*$ as $t \to \infty$.*

*Then the following four statements hold **as** $t$ **and** $L$ **tend to infinity**:*

(i) *There exist matrices $A_\infty \in \mathbb{R}^{q \times d}$ and $B_\infty \in \mathbb{R}^{d' \times q}$ such that $A^L(t)$ and $B^L(t)$ converge to $A_\infty$ and $B_\infty$.*

(ii) *There exists a Lipschitz continuous function $\mathcal{Z}_\infty : [0, 1] \to \mathbb{R}^p$ such that $Z^L_{\lfloor Ls \rfloor}(t)$ converges to $\mathcal{Z}_\infty(t)$ uniformly over $s \in [0, 1]$.*

(iii) *Uniformly over $s \in [0, 1]$ and $x \in \mathcal{X}$, the hidden layer $h^L_{\lfloor Ls \rfloor}(t)$ converges to the solution at time $s$ of the ODE*

$$
\begin{aligned}
H(0) &= A_\infty x \\
\frac{dH}{ds}(s) &= f(H(s), \mathcal{Z}_\infty(s)), \quad s \in [0, 1].
\end{aligned}
\tag{45}
$$

(iv) *Uniformly over $x \in \mathcal{X}$, the output $F^L(x; t)$ converges to $F_\infty(x) = B_\infty H(1)$. Furthermore, $F_\infty(x_i) = y_i$ for $i \in \{1, \dots, n\}$.*

*Proof.* The existence of limits $A_\infty$ and $B_\infty$ to $A^L(t)$ and $B^L(t)$ as $L$ and $t$ tend to infinity is given by Lemma 19. The same argument applies to $Z^L_{\lfloor sL \rfloor}(t)$, which provides a limit $\mathcal{Z}_\infty(s)$ to the sequence. Furthermore, following the proof of the lemma, we see that the convergence of $Z^L_{\lfloor sL \rfloor}(t)$ to $\mathcal{Z}_\infty(s)$ is uniform over $s \in [0, 1]$. Corollary 13, applied with $\theta^L_k = Z^L_k$, $\Theta_\infty = \mathcal{Z}_\infty$, $a^L = A^L x$, $g = f$, then ensures that $h^L_{\lfloor Ls \rfloor}(t)$ converges uniformly (over $s \in [0, 1]$ and $x \in \mathcal{X}$) to $H(s)$ that is the solution at time $s$ of (45), as $L$ and $t$ tend to infinity. As a consequence, $F^L(x; t)$ converges uniformly over $x$ to $F_\infty(x)$ as $L, t \to \infty$. Furthermore, recall that

$$
\ell^L(t) = \frac{1}{n} \sum_{i=1}^n \|F^L(x_i; t) - y_i\|_2^2.
$$

The left-hand side converges as $L, t \to \infty$ to $0$ by assumption of the proposition, while the right-hand side converges to

$$
\frac{1}{n} \sum_{i=1}^n \|F_\infty(x_i) - y_i\|_2^2.
$$

Therefore, $F_\infty(x_i) = y_i$ for $i \in \{1, \dots, n\}$, and the proof is complete. $\qquad \square$

# B  PROOFS OF THE RESULTS OF THE MAIN PAPER

In this section, we prove the results of the main paper. Most of these results follow from those presented in Section A. The only substantial proof is that of Proposition 5, which shows the local PL condition. It uses a result of Nguyen & Mondelli (2020) involving the Hermite transform and the sub-Gaussian variance proxy, which we define briefly. We refer to Debnath & Bhatta (2014, Chapter 17) and Vershynin (2018, Sections 2.5.2 and 3.4.1), respectively, for more detailed explanations.

**Hermite transform.**  The $r$-th normalized probabilist's Hermite polynomial is given by

$$h_r(x) = \frac{1}{\sqrt{r!}}(-1)^r e^{x^2/2}\frac{d^r}{dx^r}e^{-x^2/2}, \quad r \geqslant 0.$$

This family of polynomials forms an orthonormal basis of square-integrable functions for the inner product

$$\langle f_1, f_2 \rangle = \frac{1}{\sqrt{2\pi}}\int_{-\infty}^{\infty} f_1(x)f_2(x)e^{-x^2/2}dx.$$

Therefore, any function $\sigma$ such that $\frac{1}{\sqrt{2\pi}}\int_{-\infty}^{\infty}\sigma^2(x)e^{-x^2/2}dx < \infty$ can be decomposed on this basis. The $r$-th coefficient of this decomposition is denoted by $\eta_r(\sigma)$.

**Sub-Gaussian random vector.**  A random vector $x \in \mathbb{R}^d$ is sub-Gaussian with variance proxy $v_x > 0$ if, for every $y \in \mathbb{R}^d$ of unit norm,

$$\mathbb{P}(|\langle x, y \rangle| \geqslant t) \leqslant 2\exp\Big(-\frac{t^2}{2v_x^2}\Big).$$

**Additional notation.**  For a matrix $A$, we let $s_{\min}$ and $s_{\max}$ its minimum and maximum singular values, and similarly, $\lambda_{\min}$ and $\lambda_{\max}$ its minimum and maximum eigenvalues (whenever they exist).

Before delving into the proofs, we briefly describe the parts of this section that make use of the specific model (3). The most important one is the proof of Proposition 5, i.e., the proof that the residual network satisfies the $(M, \mu)$-local PL condition. Additionally, in the proof of Proposition 3, the expressions for $M$ and $K$ are valid only for the specific model (3). Finally, in the proof of Theorem 6, the beginning of the proof reveals that condition (22) of Proposition 8 on $\mu$ can be expressed as a condition on the norm of the labels $y_i$. This applies only to the specific model (3). Observe that, if one assumes that the general residual network of Section A satisfies the $(M, \mu)$-local PL condition with $\mu$ given by (22), then the rest of the proof of Theorem 6 unfolds, and the conclusions of the theorem hold for the general model.

## B.1  PROOF OF PROPOSITION 1

Proposition 1 is a consequence of Proposition 7 with $f(h, (V, W)) = \frac{1}{\sqrt{m}}V\sigma(\frac{1}{\sqrt{q}}Wh)$.

## B.2  PROOF OF PROPOSITION 2

Proposition 11, with $\theta_k^L = (V_k^L, W_k^L)$, $\Theta = (\mathcal{V}, \mathcal{W})$, $a^L = Ax$, $g(h, (V, W)) = \frac{1}{\sqrt{m}}V\sigma(\frac{1}{\sqrt{q}}Wh)$, gives the existence and uniqueness of the solution of the neural ODE (6). Moreover, inspecting the proof of Proposition 11, equations (35) gives that, for any input $x \in \mathcal{X}$, the difference between the last hidden layer $h_L^L$ of the discrete residual network (3) and its continuous counterpart $H(1)$ in the neural ODE (6) is bounded by

$$C'\Big(\frac{1}{L} + \sup_{s\in[0,1]}\|\Theta(s) - \Theta^L(s)\|\Big),$$

where $C' > 0$ is independent of $L$ and $x \in \mathcal{X}$, and $\Theta^L(s) = \theta^L_{\lfloor(L-1)s\rfloor+1}$. The function $\Theta^L$ is a piecewise-constant interpolation of $\Theta$ with pieces of length $\frac{1}{L-1}$. Since $\Theta$ is Lipschitz continuous, the distance between $\Theta$ and $\Theta^L$ decreases as $C''/L$ for some $C'' > 0$ depending on $\Theta$ but not on $L$. This yields $\|h_L^L - H(1)\| \leqslant \frac{C'(1++C'')}{L}$, where $C'$ and $C''$ are independent of $L$ and $x \in \mathcal{X}$. Since $F^L(x) = Bh_L^L$ and $F(x) = BH(1)$, the result is proven.

### B.3 PROOF OF PROPOSITION 3

We apply Proposition 7 with $f(h, (V, W)) = \frac{1}{\sqrt{m}} V \sigma(\frac{1}{\sqrt{q}} W h)$. Recall that the parameters $Z = (V, W)$ are considered in Proposition 7 as a vector. In particular, $\|Z\| = \|V\|_F + \|W\|_F$. Therefore, Proposition 7 shows that, for $t \in [0, T]$, $L \in \mathbb{N}^*$, and $k \in \{1, \ldots, L\}$,

$$\|A^L(t)\|_F \leqslant M, \quad \|V_k^L(t)\|_F + \|W_k^L(t)\|_F \leqslant M, \quad \text{and} \quad \|B^L(t)\|_F \leqslant M,$$

where

$$M = M_0 + T M_\pi$$
$$M_0 = \max\left(\|A^L(0)\|_F, \|V_0^L(0)\|_F + \|W_0^L(0)\|_F, \|B^L(0)\|_F\right)$$
$$M_\pi = \max\left(\max_{A \in \mathbb{R}^{q \times d}} \|\pi(A)\|_F, \max_{Z \in \mathbb{R}^{q \times m} \times \mathbb{R}^{m \times q}} \|\pi(Z)\|, \max_{B \in \mathbb{R}^{d' \times q}} \|\pi(B)\|_F\right).$$

Furthermore, due to our initialization scheme described in Section 3,

$$\|A^L(0)\|_F = \sqrt{d}, \quad \|V_0^L(0)\|_F = 0, \quad \|W_0^L(0)\|_F \leqslant 2\sqrt{qm}, \quad \|B^L(0)\|_F = \sqrt{d'},$$

where the third inequality holds with probability at least $1 - \exp(-\frac{3qm}{16})$ by Lemma 20. Since we take $q \geqslant \max(d, d')$, this implies that, with high probability, $M_0 \leqslant 2\sqrt{qm}$, yielding the formula for $M$ in Proposition 3. Finally, the existence of $K = \beta T e^{\alpha T}$ such that the difference between two successive weight matrices is bounded by $K/L$, as well as the dependence of $\alpha$ and $\beta$ on $\mathcal{X}$, $\mathcal{Y}$, $M$, and $\sigma$, follows easily from Proposition 7, given that our initialization scheme ensures that $Z_k^L(0) = Z_{k+1}^L(0)$ for all $L \in \mathbb{N}^*$ and $k \in \{1, \ldots, L\}$.

### B.4 PROOF OF THEOREM 4

By Proposition 3 and the fact that $\pi$ is bounded, the sequences $(A^L)_{L \in \mathbb{N}^*}$ and $(B^L)_{L \in \mathbb{N}^*}$ each satisfy the assumptions of Corollary 10, and $(Z_k^L)_{L \in \mathbb{N}^*, 1 \leqslant k \leqslant L}$ satisfies the assumptions of Proposition 9. Theorem 4 then follows directly from Theorem 14, by taking, as previously, $f(h, (V, W)) = \frac{1}{\sqrt{m}} V \sigma(\frac{1}{\sqrt{q}} W h)$.

### B.5 PROOF OF PROPOSITION 5

We drop the $L$ superscripts for this proof, since $L$ is fixed. Denote by $\bar{A}, \bar{B}, \bar{V}_k, \bar{W}_k$ parameters sampled according to the initialization scheme of Section 3, which means in particular that $\bar{V}_k = 0$ and $\bar{W}_k = \bar{W} \sim \mathcal{N}^{\otimes(m \times q)}$. Since, by assumption, the activation function $\sigma$ is bounded and not constant, it cannot be a polynomial function. As a consequence, there are infinitely many non-zero coefficients $\eta_r(\sigma)$ in its Hermite expansion (defined at the beginning of Section B). Throughout, we let $r \geqslant 2$ be an integer such that $\eta_r(\sigma)$ is nonzero. We also let $K_\sigma$ be the Lipschitz constant of $\sigma$ and $M_\sigma$ its supremum norm. Now, let $A, B, V_k, W_k$ be parameters at distance at most $M = \min(\frac{\eta_r(\sigma)}{32 K_\sigma \sqrt{2nq}}, \frac{1}{2})$ from $\bar{A}, \bar{B}, \bar{V}_k, \bar{W}_k$ in the sense of Definition 1.

It is useful for this proof to introduce a matrix-valued version of the residual network (3). More specifically, given data matrices $\mathbf{x} \in \mathbb{R}^{d \times n}$ and $\mathbf{y} \in \mathbb{R}^{d' \times n}$, the matrix-valued residual network writes

$$\mathbf{h}_0 = A\mathbf{x}$$
$$\mathbf{h}_{k+1} = \mathbf{h}_k + \frac{1}{L\sqrt{m}} V_{k+1} \sigma\left(\frac{1}{\sqrt{q}} W_{k+1} \mathbf{h}_k\right), \quad k \in \{0, \ldots, L-1\}, \tag{46}$$

where now $\mathbf{h}_k \in \mathbb{R}^{q \times n}$. The loss is equal to $\ell = \frac{1}{n}\|B\mathbf{h}_L - \mathbf{y}\|_F^2$ and we let $\mathbf{p}_k = \frac{\partial \ell}{\partial \mathbf{h}_k} \in \mathbb{R}^{q \times n}$ be the matrix-valued backward state. Observe that the columns of $\mathbf{x}$ are bounded and thus sub-Gaussian. In the sequel, we denote by $v_x$ the sub-Gaussian variance proxy of the columns of $\sqrt{d/q}\mathbf{x}$.

Now that we have introduced the necessary notation, we can proceed to prove some preliminary estimates. Since $M \leqslant \frac{1}{2} \leqslant \sqrt{2qm}$, we have, for $k \in \{1, \ldots, n\}$,

$$\|A - \bar{A}\|_F \leqslant M, \quad \|B - \bar{B}\|_F \leqslant \frac{1}{2}, \quad \|V_k\|_F \leqslant 1, \quad \|W_k - \bar{W}\|_F \leqslant \frac{1}{2} \leqslant \sqrt{2qm}. \tag{47}$$

By Lemma 20, with probability at least $1 - \exp\left(-\frac{qm}{16}\right)$, one has $\|\bar{W}\|_F \leqslant \sqrt{2qm}$. Together with the previous inequalities, this implies

$$\|A\|_2 \leqslant 2, \quad s_{\min}(B) \geqslant \frac{1}{2}, \quad \|B\|_2 \leqslant \frac{3}{2}, \quad \|V_k\|_F \leqslant 1, \quad \|W_k\|_F \leqslant 2\sqrt{2qm}, \qquad (48)$$

where the second inequality is a consequence of Lemma 18, as follows:

$$s_{\min}(B) \geqslant s_{\min}(\bar{B}) - \|B - \bar{B}\|_F = 1 - \|B - \bar{B}\|_F \geqslant \frac{1}{2}.$$

Let us now bound $\|\mathbf{h}_k\|_F$ and $\|\mathbf{p}_k\|_F$. We have

$$\|\mathbf{h}_0\|_F = \|A\mathbf{x}\|_F \leqslant \|A\|_2 \|\mathbf{x}\|_F \leqslant 2\sqrt{qn}. \qquad (49)$$

Moreover, by (46), for any $k \in \{0, \dots, L-1\}$,

$$\|\mathbf{h}_{k+1}\|_F \leqslant \|\mathbf{h}_k\|_F + \frac{K_\sigma}{L\sqrt{m}\sqrt{q}} \|V_{k+1}\|_F \|W_{k+1}\|_F \|\mathbf{h}_k\|_F \leqslant \left(1 + \frac{2\sqrt{2}K_\sigma}{L}\right) \|\mathbf{h}_k\|_F,$$

where the second inequality is a consequence of (48). Therefore, by (49),

$$\|\mathbf{h}_k\|_F \leqslant \exp(2\sqrt{2}K_\sigma) \|\mathbf{h}_0\|_F \leqslant 2\exp(2\sqrt{2}K_\sigma)\sqrt{qn}. \qquad (50)$$

Moving on to $\|\mathbf{p}_k\|_F$, the chain rule leads to

$$\mathbf{p}_k = \mathbf{p}_{k+1} + \frac{1}{L\sqrt{qm}} W_{k+1}^\top \left( (V_{k+1}^\top \mathbf{p}_{k+1}) \odot \sigma'\left(\frac{1}{\sqrt{q}} W_{k+1} \mathbf{h}_k\right) \right), \quad k \in \{0, \dots, L-1\},$$

where $\odot$ denotes the element-wise product. Noting that $|\sigma'| \leqslant K_\sigma$ and using (48), we obtain

$$\|\mathbf{p}_k\|_F \geqslant \|\mathbf{p}_{k+1}\|_F - \frac{K_\sigma}{L\sqrt{qm}} \|W_{k+1}\|_F \|V_{k+1}\|_F \|\mathbf{p}_{k+1}\|_F \geqslant \left(1 - \frac{2\sqrt{2}K_\sigma}{L}\right) \|\mathbf{p}_{k+1}\|_F.$$

It follows that $\|\mathbf{p}_k\|_F \geqslant \exp(-2\sqrt{2}K_\sigma) \|\mathbf{p}_L\|_F$. In addition,

$$\mathbf{p}_L = \frac{\partial \ell}{\partial \mathbf{h}_L} = \frac{2}{n} B^\top (B\mathbf{h}_L - \mathbf{y}).$$

Therefore, by Lemma 18, since $d' \leqslant q$,

$$\|\mathbf{p}_L\|_F \geqslant \frac{2}{n} s_{\min}(B) \|B\mathbf{h}_L - \mathbf{y}\|_F \geqslant \frac{1}{\sqrt{n}} \sqrt{\ell}.$$

Collecting bounds, we conclude that, for $k \in \{0, \dots, L\}$,

$$\|\mathbf{p}_k\|_F \geqslant \frac{1}{\sqrt{n}} \exp(-2\sqrt{2}K_\sigma)\sqrt{\ell}. \qquad (51)$$

A similar proof reveals that, for $k \in \{0, \dots, L\}$,

$$\|\mathbf{p}_k\|_F \leqslant \frac{3}{\sqrt{n}} \exp(2\sqrt{2}K_\sigma)\sqrt{\ell}.$$

Having established these preliminary estimates, our goal in the remainder of the proof is to lower bound the quantity $\|\frac{\partial \ell}{\partial V_{k+1}}\|_F$. First note that, by the chain rule, for any $k \in \{0, \dots, L-1\}$,

$$\frac{\partial \ell}{\partial V_{k+1}} = \frac{1}{L\sqrt{m}} \mathbf{p}_{k+1} \sigma\left(\frac{1}{\sqrt{q}} W_{k+1} \mathbf{h}_k\right)^\top.$$

As a consequence, when $m \geqslant n$, by Lemma 18,

$$\left\| \frac{\partial \ell}{\partial V_{k+1}} \right\|_F \geqslant \frac{1}{L\sqrt{m}} \|\mathbf{p}_{k+1}\|_F \cdot s_{\min}\left(\sigma\left(\frac{1}{\sqrt{q}} W_{k+1} \mathbf{h}_k\right)\right)$$

$$\geqslant \frac{1}{L\sqrt{mn}} \exp(-2\sqrt{2}K_\sigma)\sqrt{\ell} \cdot s_{\min}\left(\sigma\left(\frac{1}{\sqrt{q}} W_{k+1} \mathbf{h}_k\right)\right), \qquad (52)$$

using (51). Next, by Lemma 18,

$$s_{\min}\Big(\sigma\Big(\frac{1}{\sqrt{q}}W_{k+1}\mathbf{h}_k\Big)\Big) \geqslant s_{\min}\Big(\sigma\Big(\frac{1}{\sqrt{q}}\bar{W}\bar{A}\mathbf{x}\Big)\Big) - \Big\|\sigma\Big(\frac{1}{\sqrt{q}}W_{k+1}\mathbf{h}_k\Big) - \sigma\Big(\frac{1}{\sqrt{q}}\bar{W}\bar{A}\mathbf{x}\Big)\Big\|_F.$$

Let us first lower bound the first term. Since, by our choice of initialization, $\bar{A} = (I_{\mathbb{R}^{d\times d}}, 0_{\mathbb{R}^{(q-d)\times d}})$, we have

$$s_{\min}\Big(\sigma\Big(\frac{1}{\sqrt{q}}\bar{W}\bar{A}\mathbf{x}\Big)\Big) = s_{\min}(\sigma(\tilde{W}\tilde{\mathbf{x}})),$$

where $\tilde{W} \sim \mathcal{N}(0,1)^{\otimes(m\times d)}$ and $\tilde{\mathbf{x}} = \frac{1}{\sqrt{q}}\mathbf{x} \in \mathbb{R}^{d\times n}$ has i.i.d. unitary columns independent of $\tilde{W}$. Therefore, by Lemma 21, with probability at least $1 - \exp\big(-\frac{3m\eta_r^2(\sigma)}{64M_\sigma^2 n}\big) - 2n^2 \exp\big(-\frac{d}{2v_x n^{2/r}}\big)$,

$$s_{\min}\Big(\sigma\Big(\frac{1}{\sqrt{q}}W\bar{A}\mathbf{x}\Big)\Big) \geqslant \frac{\sqrt{m}\eta_r(\sigma)}{4}.$$

Next,

$$\Big\|\sigma\Big(\frac{1}{\sqrt{q}}W_{k+1}\mathbf{h}_k\Big) - \sigma\Big(\frac{1}{\sqrt{q}}\bar{W}\bar{A}\mathbf{x}\Big)\Big\|_F \leqslant \frac{K_\sigma}{\sqrt{q}}\Big(\|W_{k+1} - \bar{W}\|_F\|\mathbf{h}_k\|_F + \|\bar{W}\|_F\|\mathbf{h}_k - A\mathbf{x}\|_F$$
$$+ \|\bar{W}\|_F\|A\mathbf{x} - \bar{A}\mathbf{x}\|_F\Big).$$

Clearly,

$$\|\mathbf{h}_k - A\mathbf{x}\|_F = \Big\|\sum_{j=1}^k \frac{1}{L\sqrt{m}}V_j\sigma\Big(\frac{1}{\sqrt{q}}W_j\mathbf{h}_{j-1}\Big)\Big\|_F \leqslant \frac{4\sqrt{2}K_\sigma k}{L}\exp(2\sqrt{2}K_\sigma)\sqrt{qn},$$

by (48) and (50). Also,

$$\|A\mathbf{x} - \bar{A}\mathbf{x}\|_F \leqslant \|A - \bar{A}\|_F\|\mathbf{x}\|_F \leqslant \frac{\eta_r(\sigma)}{32\sqrt{2}K_\sigma},$$

by (47) and by definition of $M$. Putting together the two bounds above as well as (47), (48), and (50), we obtain

$$\Big\|\sigma\Big(\frac{1}{\sqrt{q}}W_{k+1}\mathbf{h}_k\Big) - \sigma\Big(\frac{1}{\sqrt{q}}W\bar{A}\mathbf{x}\Big)\Big\|_F \leqslant K_\sigma\exp(2\sqrt{2}K_\sigma)\sqrt{n}\Big(1 + \sqrt{qm}\frac{8K_\sigma k}{L}\Big) + \sqrt{m}\frac{\eta_r(\sigma)}{32}$$
$$\leqslant C_1\sqrt{n} + C_2\frac{\sqrt{nqm}k}{16L} + \sqrt{m}\frac{\eta_r(\sigma)}{32},$$

where $C_1 = K_\sigma\exp(2\sqrt{2}K_\sigma)$ and $C_2 = 128C_1K_\sigma$. Thus, when $C_1\sqrt{n} \leqslant \frac{1}{32}\sqrt{m}\eta_r(\sigma)$, we have

$$s_{\min}\Big(\sigma\Big(\frac{1}{\sqrt{q}}W_{k+1}\mathbf{h}_k\Big)\Big) \geqslant \sqrt{m}\Big(\frac{3}{16}\eta_r(\sigma) - \frac{C_2}{16}\sqrt{nq}\frac{k}{L}\Big) \geqslant \frac{1}{8}\sqrt{m}\eta_r(\sigma)$$

for $k \leqslant \frac{L\eta_r(\sigma)}{C_2\sqrt{nq}}$. As a consequence, for $k \leqslant \frac{L\eta_r(\sigma)}{C_2\sqrt{nq}}$, returning to (52),

$$\Big\|\frac{\partial\ell}{\partial V_{k+1}}\Big\|_F \geqslant \frac{1}{8L\sqrt{n}}\eta_r(\sigma)\exp(-2\sqrt{2}K_\sigma)\sqrt{\ell} = \frac{C_3\eta_r(\sigma)}{L\sqrt{n}}\sqrt{\ell},$$

letting $C_3 = \frac{\exp(-2\sqrt{2}K_\sigma)}{8}$. Therefore,

$$\Big\|\frac{\partial\ell}{\partial A}\Big\|_F^2 + L\sum_{k=1}^L\Big\|\frac{\partial\ell}{\partial Z_{k+1}}\Big\|_F^2 + \Big\|\frac{\partial\ell}{\partial B}\Big\|_F^2 \geqslant L\sum_{k=1}^{\lfloor\frac{L\eta_r(\sigma)}{C_2\sqrt{nq}}\rfloor}\Big\|\frac{\partial\ell}{\partial V_{k+1}}\Big\|_F^2$$
$$\geqslant L\Big\lfloor\frac{L\eta_r(\sigma)}{C_2\sqrt{nq}}\Big\rfloor\frac{C_3^2\eta_r(\sigma)^2}{L^2 n}\ell$$
$$\geqslant \frac{C_3^2\eta_r(\sigma)^3}{2C_2 n\sqrt{nq}}\ell,$$

where we used the inequality $\lfloor x \rfloor \geqslant x/2$ for $x \geqslant 1$. This proves the result, with

$$c_1 = \max\left(\frac{2^{10}C_1^2}{\eta_r(\sigma)^2}, 1\right) = \max\left(\frac{2^{10}K_\sigma^2 \exp(4\sqrt{2}K_\sigma)}{\eta_r(\sigma)^2}, 1\right)$$

$$c_2 = \frac{C_2}{\eta_r(\sigma)} = \frac{128K_\sigma^2 \exp(2\sqrt{2}K_\sigma)}{\eta_r(\sigma)}$$

$$c_3 = \min\left(\frac{\eta_r(\sigma)}{32\sqrt{2}K_\sigma}, \frac{1}{2}\right)$$

$$c_4 = \frac{C_3^2 \eta_r(\sigma)^3}{2C_2} = \frac{\eta_r(\sigma)^3}{2^{14}K_\sigma^2 \exp(6\sqrt{2}K_\sigma)}$$

$$\delta = \exp\left(-\frac{qm}{16}\right) + n\exp\left(-\frac{3m\eta_r^2(\sigma)}{64M_\sigma^2 n}\right) + 2n^2 \exp\left(-\frac{d}{2v_x n^{2/r}}\right).$$

**Remark 2.** *With appropriate values of $r$ and $m$, the probability of failure $\delta$ can be made as small as*

$$\varepsilon + 2n^2 \exp\left(-\frac{d}{2v_x n^\varepsilon}\right), \tag{53}$$

*for any $\varepsilon > 0$. This is possible first by choosing $r$ such that $2/r \geqslant \varepsilon$, then by choosing $m$ such that the first two terms are less than $\varepsilon$. Moreover, we refer the interested reader to Goel et al. (2020, Lemmas A.2 and A.9) for quantitative estimates of $\eta_r(\sigma)$ for ReLU and sigmoid activations. Finally, the expression (53) is essentially the same as the one appearing in Nguyen & Mondelli (2020, Theorem 3.3). As in this paper, we note that this expression is small if $n$ grows at most polynomially with $d$, in which case the exponential term in $d$ dominates the polynomial term in $n$.*

### B.6 PROOF OF THEOREM 6

By Proposition 5, there exists $\delta > 0$ such that, with probability at least $1 - \delta$, the residual network (3) satisfies the $(M, \mu)$-local PL condition around its initialization, with

$$M = \frac{c_3}{\sqrt{nq}} \quad \text{and} \quad \mu = \frac{c_4}{n\sqrt{nq}},$$

for $c_3$ and $c_4$ depending on $\sigma$. Let us now apply Proposition 8 with $f(h, (V, W)) = \frac{1}{\sqrt{m}}V\sigma(\frac{1}{\sqrt{q}}Wh)$. The only assumption of Proposition 8 that requires some care to check is that the PL condition holds for the value of $\mu$ given by equation (22). Since the $(M, \mu)$-local PL condition implies the $(M, \tilde{\mu})$-local PL condition for any $\tilde{\mu} \in (0, \mu)$, it is the case if

$$\frac{c_4}{n\sqrt{nq}} \geqslant \max(M_B K, M_B M_X, M_A M_X)\frac{8e^K}{M}\sup_{L\in\mathbb{N}^*}\sqrt{\ell^L(0)},$$

with $M_X$, $M_A$, $M_B$, and $K$ defined in Proposition 8. Due to the initialization scheme of Section 3, we have, for any input $x \in \mathcal{X}$, $h_L^L(0) = h_0^L(0)$, hence $F^L(x) = B^L(0)A^L(0)x = 0$ since $q \geqslant d + d'$. As a consequence, $\ell^L(0) = \frac{1}{n}\sum_{i=1}^{n}\|y_i\|^2$. Therefore, the condition becomes

$$\frac{1}{n}\sum_{i=1}^{n}\|y_i\|^2 \leqslant \frac{c_3^2 c_4^2}{64n^4 q^3 \max(M_B K, M_B M_X, M_A M_X)^2 e^{2K}},$$

where we replaced $M$ by its value. Define $C$ to be equal to the constant on the right-hand side. Then, according to the above, as soon as $\frac{1}{n}\sum_{i=1}^{n}\|y_i\|^2 \leqslant C$, we can apply Proposition 8, which gives several guarantees. First, the gradient flow is well defined on $\mathbb{R}_+$. Moreover, the proposition and the expression of $\mu$ given above yield the bound on the empirical risk. In particular, the empirical risk converges uniformly to zero. Furthermore, Proposition 8 shows the uniform convergence of the weights as $t \to \infty$. Finally, the proposition ensures that the sequences $(A^L)_{L\in\mathbb{N}^*}$ and $(B^L)_{L\in\mathbb{N}^*}$ each satisfy the assumptions of Corollary 10, and that $(Z_k^L)_{L\in\mathbb{N}^*, 1\leqslant k\leqslant L}$ satisfies the assumptions of Proposition 9. We can therefore apply Theorem 14, with $f$ defined above and $\pi$ equal to the identity. This gives the uniform convergence of the weights as $L \to \infty$. The four asymptotic statements of Theorem 6 are then a consequence of Proposition 15.

**Remark 3.** *A close examination of the quantities involved in the definition of $C$ reveals that it depends only on $\mathcal{X}$, $\sigma$, $n$, and $q$. In particular, it does not depend on the dimension $m$.*

## C    SOME TECHNICAL LEMMAS

We start by recalling the Picard-Lindelöf theorem (see, e.g., Luk, 2017, for a self-contained presentation, and Arnold, 1992, for a textbook).

**Lemma 16** (Picard-Lindelöf theorem). *Let $I = [0, T] \subset \mathbb{R}_+$ be an interval, for some $T \in (0, \infty]$. Consider the initial value problem*

$$U(s) = U_0 + \int_0^s g(U(r), r)dr, \quad s \in I, \tag{54}$$

*where $g : \mathbb{R}^d \times I \to \mathbb{R}^d$ is continuous and locally Lipschitz continuous in its first variable. Then the initial value problem is well defined on an interval $[0, T_{\max}) \subset I$, i.e., there exists a unique maximal solution on this interval. Moreover, if $T_{\max} < T$, then $\|U(s)\|$ tends to infinity when $s$ tends to $T_{\max}$. Finally, if $g(\cdot, r)$ is uniformly Lipschitz continuous for $r$ in any compact, then $T_{\max} = T$.*

We define time-dependent dynamics (54) for generality, but the time-independent case $U(s) = U_0 + \int_0^s g(U(r))dr$ is also of interest. In this case, the existence and uniqueness of the maximal solution holds if $g$ is locally Lipschitz continuous, and the solution is defined on $I$ if $g$ is Lipschitz continuous. Besides, the first statement of Lemma 16 (existence and uniqueness of the maximal solution) also holds if $\mathbb{R}^d$ is replaced by any (potentially infinite-dimensional) Banach space.

The next lemma gives conditions for the existence and uniqueness of the global solution of the initial value problem (54) when the assumption of continuity of $g$ in its second variable is removed, thereby generalizing the Picard-Lindelöf theorem.

**Lemma 17** (Caratheodory conditions for the existence and uniqueness of the global solution of an initial value problem). *Consider the initial value problem*

$$U(s) = U_0 + \int_0^s g(U(r), r)dr, \quad s \in [0, 1],$$

*where $g : \mathbb{R}^d \times [0, 1] \to \mathbb{R}^d$ is measurable and the integral is understood in the sense of Lebesgue integration. Assume that $g(\cdot, r)$ is uniformly Lipschitz continuous for almost all $r \in [0, 1]$, and that $g(0, r) \equiv 0$. Then there exists a unique solution to the initial value problem, defined on $[0, 1]$.*

*Proof.* The proof is a consequence of Filippov (1988, Theorems 1, 2, and 4). More specifically, denote by $C > 0$ the uniform Lipschitz constant of $g(\cdot, r)$. According to Filippov (1988, Theorems 1 and 2), under the conditions of the lemma, there exists a unique maximal solution to the initial value problem. Let us now consider a restricted version of the problem, where $g$ is defined on $D \times [0, 1]$, with $D$ a compact of $\mathbb{R}^d$ large enough to contain in its interior the ball of center 0 and radius $\|U_0\| \exp(C)$. There exists a unique maximal solution to this problem as well, also according to Filippov (1988, Theorems 1 and 2), and, according to Filippov (1988, Theorem 4), it is defined until it reaches the boundary of $D \times [0, 1]$, which it reaches at some point $(U^*, s^*)$. If $s^* < 1$, it means that $U^*$ is on the boundary of $D$, and in particular that $\|U^*\| > \|U_0\| \exp(C)$. But, on the other hand, for almost every $r \in [0, 1]$,

$$\|g(U(r), r)\| \leqslant \|g(0, r)\| + \|g(U(r), r) - g(0, r)\| \leqslant C\|U(r)\|.$$

Hence, by Grönwall's inequality, for $s \leqslant s^*$,

$$\|U(s)\| \leqslant \|U_0\| \exp(C).$$

Thus, $\|U^*\| \leqslant \|U_0\| \exp(C)$, which is impossible. Hence the maximal solution of the restricted problem is defined on $[0, 1]$. Furthermore, the maximal solution of the original problem coincides with the restricted one whenever $U(s) \in D$, which is the case for every $s \in [0, 1]$, hence the maximal solution is defined on $[0, 1]$. $\qquad\square$

The next three lemmas recall well-known results from linear algebra, analysis, and random matrix theory. Recall that $s_{\min}$ and $\lambda_{\min}$ denote respectively the minimum singular value and eigenvalue of a matrix.

**Lemma 18.** *Let $A, A' \in \mathbb{R}^{m \times r}$ and $B \in \mathbb{R}^{r \times n}$. Then*

$$s_{\min}(A + A') \geqslant s_{\min}(A) - \|A'\|_F.$$

*If $m \geqslant r$, then $\|AB\|_F \geqslant s_{\min}(A)\|B\|_F$. Furthermore, if $n \geqslant r$, then $\|AB\|_F \geqslant \|A\|_F s_{\min}(B)$.*

*Proof.* The first statement is a consequence of, e.g., Loyka (2015), which establishes that $s_{\min}(A + A') \geqslant s_{\min}(A) - s_{\max}(A')$, yielding the first inequality since $s_{\max}(A') = \|A\|_2 \leqslant \|A\|_F$. As for the second one, we have

$$\|AB\|_F^2 = \text{Tr}(ABB^\top A^\top) = \text{Tr}(BB^\top A^\top A) \geqslant \lambda_{\min}(A^\top A)\text{Tr}(BB^\top) = \lambda_{\min}(A^\top A)\|B\|_F^2.$$

Since $m \geqslant r$, the rightmost quantity is equal to $s_{\min}(A)\|B\|_F$, proving the second statement of the lemma. The third statement is similar. $\square$

**Lemma 19.** *Let $(e_{x,y})_{x \in \mathbb{R}_+, y \in \mathbb{R}_+} \subset E$, where $E$ is a Banach space, such that $e_{x,y}$ converges uniformly to $e_{\infty,y}$ when $x \to \infty$, and converges uniformly to $e_{x,\infty}$ when $y \to \infty$. Then there exists $e_\infty \in E$ such that*

$$\lim_{x,y \to \infty} e_{x,y} = \lim_{x \to \infty} e_{x,\infty} = \lim_{y \to \infty} e_{\infty,y} = e_\infty.$$

*Proof.* Let $\varepsilon > 0$. Since $e_{x,y}$ converges uniformly to $e_{\infty,y}$ as $x \to \infty$, there exists $x_0 \in \mathbb{R}_+$ such that, for $x_1, x_2 > x_0$ and $y \in \mathbb{R}_+$,

$$\|e_{x_1,y} - e_{x_2,y}\| \leqslant \frac{\varepsilon}{2}.$$

Similarly, there exists $y_0 \in \mathbb{R}_+$ such that, for $x \in \mathbb{R}_+$ and $y_1, y_2 > y_0$,

$$\|e_{x,y_1} - e_{x,y_2}\| \leqslant \frac{\varepsilon}{2}.$$

Hence, for $x_1, x_2 > x_0$ and $y_1, y_2 > y_0$,

$$\|e_{x_1,y_1} - e_{x_2,y_2}\| \leqslant \|e_{x_1,y_1} - e_{x_1,y_2}\| + \|e_{x_1,y_2} - e_{x_2,y_2}\| \leqslant \varepsilon.$$

We conclude that $(e_{x,y})_{x \in \mathbb{R}_+, y \in \mathbb{R}_+}$ is a Cauchy sequence, which therefore converges to some limit $e_\infty \in E$. $\square$

**Lemma 20.** *Let $W \in \mathbb{R}^{q \times m}$ be a standard Gaussian random matrix. Then, for $M_W \geqslant \sqrt{2}$, with probability at least $1 - \exp(-\frac{(M_W^2 - 1)qm}{16})$, one has $\|W\|_F \leqslant M_W\sqrt{q}\sqrt{m}$.*

*Proof.* The quantity $\|W\|_F^2$ follows a chi-squared distribution with $qm$ degrees of freedom. Hence, according to Laurent & Massart (2000, Lemma 1), for $x \geqslant 0$,

$$\mathbb{P}(\|W\|_F^2 - qm \geqslant 2\sqrt{qmx} + 2x) \leqslant \exp(-x).$$

Taking $x = \frac{(M_W^2 - 1)qm}{16}$, we see that

$$2\sqrt{qmx} = \frac{1}{2}\sqrt{M_W^2 - 1}qm \leqslant \frac{1}{2}(M_W^2 - 1)qm,$$

where the bound follows from $M_W \geqslant \sqrt{2}$. Since furthermore $2x \leqslant \frac{1}{2}(M_W^2 - 1)qm$, we obtain

$$2\sqrt{qmx} + 2x \leqslant (M_W^2 - 1)qm,$$

and thus

$$\mathbb{P}(\|W\|_F^2 > M_W^2 qm) \leqslant \mathbb{P}(\|W\|_F^2 - qm \geqslant 2\sqrt{qmx} + 2x) \leqslant \exp(-x),$$

yielding the result. $\square$

Finally, the last lemma of the section gives a lower bound on the smallest singular value of a matrix of the form $\sigma(A)$, where $\sigma$ is a bounded function applied element-wise and $A$ belongs to a family of random matrix. The lower bound involves the Hermite transform of $\sigma$, which is defined in Section B.

**Lemma 21.** *Let $\sigma$ be a function bounded by some $M_\sigma > 0$. Let $W \in \mathbb{R}^{m \times d}$ be a standard Gaussian random matrix, and $X \in \mathbb{R}^{d \times n}$ a random matrix with i.i.d. unitary columns independent of $W$. Then, for any integer $r \geqslant 2$, there exists $\delta > 0$ such that, with probability at least $1 - \delta$, the smallest singular value of $\sigma(WX)$ is greater than $\frac{1}{4}\sqrt{m}\eta_r(\sigma)$, where $\eta_r(\sigma)$ is the $r$-th coefficient in the Hermite transform of $\sigma$. Furthermore, the following expression for $\delta$ holds:*

$$\delta = n\exp\Big(-\frac{3m\eta_r^2(\sigma)}{64M_\sigma^2 n}\Big) + 2n^2\exp\Big(-\frac{d}{2Cn^{2/r}}\Big),$$

*where $C$ is the sub-Gaussian variance proxy of the columns of $\sqrt{d}X$.*

*Proof.* Denoting by $w_i$ the $i$-th row of $W$ and letting

$$M_i = \sigma(X^\top w_i^\top)\sigma(w_i X),$$

our goal is to lower bound the smallest eigenvalue value $\lambda_{\min}(M)$ of $M = \sum_{i=1}^m M_i$. Observe that

$$\mathbb{E}(M|X) = m\mathbb{E}_{\tilde{w}\sim\mathcal{N}(0,I_d)}\Big(\sigma(X^\top \tilde{w}^\top)\sigma(\tilde{w}X)\Big|X\Big)$$

$$= m\mathbb{E}_{\tilde{w}\sim\mathcal{N}(0,\frac{1}{d}I_d)}\Big(\sigma\big((\sqrt{d}X)^\top \tilde{w}^\top\big)\sigma(\tilde{w}(\sqrt{d}X))\Big|X\Big).$$

Letting $\lambda_{\min}(\mathbb{E}(M|X))$ be the smallest eigenvalue of this matrix and $r \geqslant 2$ be an integer, Nguyen & Mondelli (2020, Lemma 3.4) show that, with probability at least $1 - 2n^2 \exp(-\frac{d}{2Cn^{2/r}})$ over the matrix $X$,

$$\lambda_{\min}(\mathbb{E}(M|X)) \geqslant \frac{m\eta_r^2(\sigma)}{8}. \tag{55}$$

We now apply a matrix Chernoff's bound to lower bound with high probability the smallest eigenvalue $\lambda_{\min}(M|X)$ of $M$ conditionally on $X$, as a function of $\lambda_{\min}(\mathbb{E}(M|X))$. By Tropp (2012, Remark 5.3), we have, for $t \in [0,1]$,

$$\mathbb{P}(\lambda_{\min}(M) \leqslant t\lambda_{\min}(\mathbb{E}(M|X))|X) \leqslant n\exp\Big(-\frac{(1-t^2)\lambda_{\min}(\mathbb{E}(M|X))}{2R(X)}\Big),$$

where $R(X)$ is an almost sure upper bound on the largest eigenvalue of $M_i|X$, which we can take equal to $M_\sigma^2 n$ since the largest eigenvalue of $M_i$ is equal to $\|\sigma(w_i X)\|_2^2 \leqslant M_\sigma^2 n$. Taking $t = 1/2$, we obtain, on the event $[\lambda_{\min}(\mathbb{E}(M|X)) \geqslant \frac{m\eta_r^2(\sigma)}{8}]$,

$$\mathbb{P}\Big(\lambda_{\min}(M) \geqslant \frac{\lambda_{\min}(\mathbb{E}(M|X))}{2}\Big|X\Big) \geqslant 1 - n\exp\Big(-\frac{3m\eta_r^2(\sigma)}{64M_\sigma^2 n}\Big),$$

thus, on the event $[\lambda_{\min}(\mathbb{E}(M|X)) \geqslant \frac{m\eta_r^2(\sigma)}{8}]$,

$$\mathbb{P}\Big(\lambda_{\min}(M) \geqslant \frac{m\eta_r^2(\sigma)}{16}\Big) \geqslant 1 - n\exp\Big(-\frac{3m\eta_r^2(\sigma)}{64M_\sigma^2 n}\Big).$$

Using (55), we obtain

$$\mathbb{P}\Big(\lambda_{\min}(M) \geqslant \frac{m\eta_r^2(\sigma)}{16}\Big) \geqslant \Big(1 - n\exp\Big(-\frac{3m\eta_r^2(\sigma)}{64M_\sigma^2 n}\Big)\Big)\mathbb{P}\Big(\lambda_{\min}(\mathbb{E}(M|X)) \geqslant \frac{m\eta_r^2(\sigma)}{8}\Big)$$

$$\geqslant \Big(1 - n\exp\Big(-\frac{3m\eta_r^2(\sigma)}{64M_\sigma^2 n}\Big)\Big)\Big(1 - 2n^2\exp\Big(-\frac{d}{Cn^{2/r}}\Big)\Big)$$

$$\geqslant 1 - n\exp\Big(-\frac{3m\eta_r^2(\sigma)}{64M_\sigma^2 n}\Big) - 2n^2\exp\Big(-\frac{d}{2Cn^{2/r}}\Big).$$

$\square$

## D   COUNTER-EXAMPLE FOR THE RELU CASE.

This section gives a proof sketch to illustrate that, with the ReLU activation $\sigma : x \mapsto \max(0, x)$, the smoothness of the weights can be lost during training. More precisely, we show a case where successive weights are at distance $\mathcal{O}(\frac{1}{L})$ at initialization and at distance $\Omega(1)$ after training.

For the sake of simplicity, we will assume that the depth is even, and denote it as $2L$. We place ourselves in a one-dimensional setting (i.e., $d = 1$). The parameters are $(w_1, \cdots, w_{2L}) \in \mathbb{R}^{2L}$, and the residual network writes as follows, for an input $x \in \mathbb{R}$:

$$h_0(t) = x$$

$$h_{k+1}(t) = h_k(t) + \frac{1}{2L}\sigma(w_{k+1}(t)h_k(t)), \quad k \in \{0, \dots, 2L - 1\}.$$

We consider a sample consisting of a single point $(x, Cx) \in \mathbb{R}_+^2$, with $C > 1$ (independent of $L$), and define the empirical risk as $\ell(t) = (h_{2L}(t) - Cx)^2$. The risk is minimized by gradient flow.

The weights are initialized to $w_k(0) = \frac{(-1)^k}{2L}$. For $x \in \mathbb{R}_+$ we have that $h_k(t) \geqslant 0$ for all $k \in \{0, \ldots, 2L\}$. Note that the argument of $\sigma$ on the odd layers is negative. Therefore, by definition of $\sigma$, the gradient of the loss with respect to the odd layers is zero and we have, for $k \in \{0, \ldots, L-1\}$, $w_{2k+1}(t) = w_{2k+1}(0)$. On the other hand, the argument of $\sigma$ is positive on the even layers, and thus,

$$h_{2L}(t) = \prod_{j=1}^{L} \Big(1 + \frac{w_{2j}(t)}{2L}\Big) x.$$

As a consequence, the gradient flow equation for the even layers is, for $k \in \{1, \ldots, L\}$,

$$\frac{dw_{2k}}{dt}(t) = -\frac{\partial \ell}{\partial w_{2k}}(t) = 2x \Big(C - \prod_{j=1}^{L} \Big(1 + \frac{w_{2j}(t)}{2L}\Big)\Big) \prod_{j=1, j \neq k}^{L} \Big(1 + \frac{w_{2j}(t)}{2L}\Big).$$

Due to the symmetry of these equations for $k \in \{1, \ldots, L\}$ and the fact that all the $w_{2k}(0)$ are equal, the parameters on each even layer coincide at all times and are equal to $w(t)$ such that

$$\frac{dw}{dt}(t) = 2x \Big(C - \Big(1 + \frac{w(t)}{2L}\Big)^L\Big) \Big(1 + \frac{w(t)}{2L}\Big)^{L-1}.$$

An analysis of this ODE reveals that $w(t)$ tends as $t \to \infty$ to $w^\star > 0$ satisfying that

$$\Big(1 + \frac{w^\star}{2L}\Big)^L = C. \tag{56}$$

This can be seen by letting $y(t) = C - (1 + \frac{w(t)}{2L})^L$, and applying Grönwall's inequality to $y$. Therefore, as $t \to \infty$, one has $w_{2k+1}(t) \to -\frac{1}{2L}$ and $w_{2k}(t) \to w^\star$, where (56) implies that $w^\star \geqslant 2 \log(C)$. This shows that the final weights are not smooth in the sense that the distance between two successive weights is $\Omega(1)$.

This result contrasts sharply with Proposition 7, which shows that successive weights remain at a distance $\mathcal{O}(\frac{1}{L})$ throughout training, when initialized as a discretization of a Lipschitz continuous function, and with a smooth activation function. In fact, Proposition 7 can be generalized to any initialization such that successive weights are at distance $\mathcal{O}(\frac{1}{L})$ at initialization, which is the case in the counter-example. This means that the only broken assumption in our counter-example is the non-smoothness of the activation function. This non-smoothness causes the gradient flow dynamics for two successive weights to deviate, even though the weights are initially close to each other, because they are separated by the kink of ReLU at zero.

# E  EXPERIMENTAL DETAILS

Our code is available at https://github.com/michaelsdr/implicit-regularization-resnets-nodes.

We use Pytorch (Paszke et al., 2019).

**Synthetic data.** To ease the presentation, we consider the case where $q = d = d'$, and we do not train the weights $A^L$ and $B^L$, which therefore stay equal to the identity. The sample points $(x_i, y_i)_{1 \leqslant i \leqslant n}$ follow independent standard Gaussian distributions. Note that it does not hurt to take $x$ and $y$ independent since, in this subsection, our focus is on optimization results only and not on statistical aspects.

**Large-depth limit.** We take $n = 100$, $d = 16$, $m = 32$. We train for 500 iterations, and set the learning rate to $L \times 10^{-2}$. The scaling of the learning rate with $L$ is the equivalent of the $L$ factor in the gradient flow (4).

**Long-time limit.** We take $n = 50$, $d = 16$, $m = 64$, $L = 64$, and train for 80,000 iterations with a learning rate of $5L \times 10^{-3}$.

**Real-world data.** We take $L = 256$. The first layer is a trainable convolutional layer with a kernel size of $5 \times 5$, a stride of 2, a padding of 1, and 16 out channels. We then iterate the residual layers

$$h_{k+1}^L = h_k^L + \frac{1}{L}\mathrm{bn}_{2,k}^L(\mathrm{conv}_{2,k}^L(\sigma(\mathrm{bn}_{1,k}^L(\mathrm{conv}_{1,k}^L(h_k^L))))), \quad k \in \{0, \ldots, L-1\},$$

where $\mathrm{conv}_{i,k}^L$ are convolutions with kernel size 3, stride of 2, and padding of 1, and $\mathrm{bn}_{i,k}^L$ are batch normalizations, as is standard in residual networks (He et al., 2016b). The model is trained using stochastic gradient descent on the cross-entropy loss for 180 epochs. The initial learning rate is $4 \times 10^{-2}$ and is gradually decreased using a cosine learning rate scheduler.

**Normalization.** The residual layers considered in the real-world case have a batch normalization layer (see formula above). We observe empirically that implicit regularization towards a neural ODE still holds in this case. However, these layers are not present in the models we consider. Nevertheless, as discussed in Section 4.3, some of our results extend to a setting where we only assume that the residual connection is a Lipschitz-continuous function. The intuition suggests that this should include in particular the case where layer normalizations are added to the architecture, although this should clearly necessitate a rigorous and separate mathematical analysis. Finally, note that a connection has been drawn between batch normalization and scaling factors (De & Smith, 2020).

**Additional plot.** To complement Figure 3, we display the average (across layers) of the Frobenius norm of the difference between two successive weights in the convolutional ResNets after training on CIFAR-10, depending on the initialization strategy. The index $i$ corresponds to the index of the convolution layer. Results are averaged over 5 runs. We see that a smooth initialization leads to weights that are in average an order of magnitude smoother than those obtained with an i.i.d. initialization.

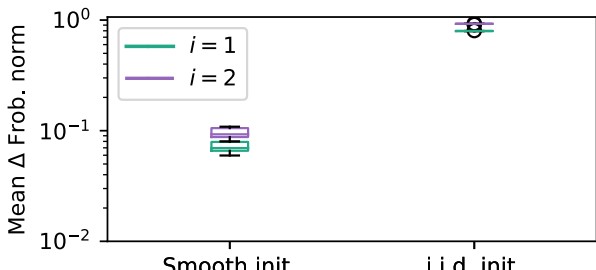

Figure 4: Average (across layers) of the Frobenius norm of the difference between two successive weights in the convolutional ResNets after training on CIFAR-10, depending on the initialization strategy.

