# OpenReview forum: "Implicit regularization of deep residual networks towards neural ODEs"
_ICLR.cc/2024/Conference — ICLR 2024 spotlight_

### Official Review · Reviewer_Htsq · 2023-10-30

**Soundness:** 4 excellent
**Presentation:** 4 excellent
**Contribution:** 4 excellent
**Rating:** 8
**Confidence:** 4

**Summary:**

The paper establishes that initialising the network with an ODE results in preserving smoothness of weights throughout training.  The authors do this through the framework of Polyak-Łojasiewicz condition.

**Strengths:**

Pros:

- (Significance) It is an important result that shows the impact of initial conditions on the training of ResNets; furthermore, it is crucial that while the paper is mostly theoretical, the settings the authors propose are realistic and can generalise to the practical ResNet models.  I also checked the derivations and could not find any problems with them.

- (Quality) The paper is well written, with thoroughly addressing the reproducibility aspects (through both derivations and the experiments)

- (Clarity) The paper is clearly written, with great attention to detail

- (Originality) While the notion of interpretation of ResNets as a discretisation of (Neural) ODEs has been a widely-discussed topic, it has not been analysed in the opposite direction. The authors address an intriguing question of whether every ResNet corresponds to a (Neural) ODE and show, to my knowledge, previously undocumented, impact of initial conditions on whether the ResNet can be presented as a discretisation of an ODEs

**Weaknesses:**

Cons:

-  (Significance elements) There are some limitations to the analysis, which are related to implicit generalisation to the convolutional and other sparse scenarios (see questions below).

**Questions:**

Q 1: It seems to me that both fully-connected and convolutional neural networks would be the particular cases where the ODE parameter matrices A, V, W are considered sparse (some or most of the entries are fixed to zero and the rest are optimised according to Equation (5)). The experiment, depicted in Figure 3, suggests that results in Theorem 4 might be expanded to the more generic sparse scenarios.  Might be good if the authors could confirm this observation.

Q2: If it happens that the weights are initialised randomly, for every finite depth of the neural network (as there’s finite time and finite depth), there could be still a Lipschitz-continuous function (ii) over that set (as locally Lipschitz function on a compact set is Lipschitz as per, e.g. https://faculty.etsu.edu/gardnerr/5510/cspace.pdf ). However, what’s appearing in Figure 3 appears to go beyond literal application of Theorem 4: not only the function is compact but also it appears that the Lipshitz constant is preserved to be small during the optimisation.

Q3: Another interesting aspect is that  it would be intuitive to see smaller Lipshitz constant could be beneficial for out-of-distribution recognition. I wonder if the authors could clarify on this aspect? To what extent would the proposed derivation generalise for higher-than-two layers of multilayer perceptrons?

---

> ### Author Response · Authors · 2023-11-15
>
> > Q 1: It seems to me that both fully-connected and convolutional neural networks would be the particular cases where the ODE parameter matrices A, V, W are considered sparse (some or most of the entries are fixed to zero and the rest are optimised according to Equation (5)). The experiment, depicted in Figure 3, suggests that results in Theorem 4 might be expanded to the more generic sparse scenarios. Might be good if the authors could confirm this observation.
>
> We confirm that the reviewer is right. In Section 4.3, we describe a general setup where Theorem 4 holds, where it is only assumed that the residual connection $f$ is a Lipschitz-continuous $C^2$ function. This includes in particular the sparse version of the model considered in the paper. This is an interesting comment that we will incorporate in the next version of the paper.
>
> > Q2: If it happens that the weights are initialized randomly, for every finite depth of the neural network (as there’s finite time and finite depth), there could be still a Lipschitz-continuous function (ii) over that set (as locally Lipschitz function on a compact set is Lipschitz as per, e.g. https://faculty.etsu.edu/gardnerr/5510/cspace.pdf ). However, what’s appearing in Figure 3 appears to go beyond literal application of Theorem 4: not only the function is compact but also it appears that the Lipshitz constant is preserved to be small during the optimisation.
>
> This is a very good point, thanks to the reviewer for raising it. It is true that for a fixed depth, it is always possible to construct a Lipschitz-continuous function that interpolates the weights, e.g., by piecewise-affine interpolation. Nevertheless, without further assumption, the Lipschitz constant of such a function depends on the depth $L$. On the contrary, as observed by the reviewer, in our results, the Lipschitz constant of the function that interpolates the weights is independent of the depth (see, e.g., Theorem 4 (ii) and Theorem 6 (ii)). Moreover, the proofs give a bound on this Lipschitz constant during training, even though we did not state it in the theorems to simplify the exposition. We will add a remark about this in the next version of the paper.
>
> > Q3: Another interesting aspect is that it would be intuitive to see smaller Lipshitz constant could be beneficial for out-of-distribution recognition. I wonder if the authors could clarify on this aspect?
>
> Yes, this is a correct intuition. This is confirmed by a recent NeurIPS paper (Marion 2023, ref. below), where it is shown that the generalization error of deep residual networks / neural ODEs grows with the Lipschitz constant.
>
> Generalization bounds for neural ordinary differential equations and deep residual networks, Marion, NeurIPS 2023.
>
> > To what extent would the proposed derivation generalise for higher-than-two layers of multilayer perceptrons?
>
> As mentioned above, Section 4.3 presents generalizations of our results to an arbitrary residual connection $f$, which we assume to be a Lipschitz-continuous $C^2$ function. This includes in particular the case where $f$ is a deeper MLP. Our finite training time results are fully proven in this general case. Regarding the limit $T \to \infty$, we prove that convergence to a neural ODE holds for a general residual connection $f$ if it satisfies a Polyak-Lojasiewicz (PL) condition. Proving the PL condition in the case where the residual connection is a deeper MLP is a challenging mathematical question, which deserves future work.

---

> ### Comment · Reviewer_Htsq · 2023-11-20
> **Response to the rebuttal**
>
> Dear Authors,
>
> I've gone through  the responses to my review and to the ones by other reviewers. I think the questions are sufficiently answered, and my score remains unchanged.
>
> Many thanks,
> The Reviewer

---

### Official Review · Reviewer_Hmrt · 2023-10-31

**Soundness:** 3 good
**Presentation:** 3 good
**Contribution:** 3 good
**Rating:** 6
**Confidence:** 2

**Summary:**

The paper establishes a connection between deep residual networks and neural odes.  Prior works have attempted to provide a theoretical framework for the same, but they come with several shortcomings -- like considered only simplified models with linear activations. The authors train the neural network with gradient flow and show that in the limit of $L \rightarrow \infty$, the network converges to a discretization of neural ode.

**Strengths:**

1. Most prior works in this domain make simplifying assumptions which prevents their analysis from being directly applicable to practical models. The authors overcome those assumptions

**Weaknesses:**

N/A

I am giving a score of 6, given my inability to appropriately verify all the proofs in the main text and appendices.

**Questions:**

N/A

---

> ### Author Response · Authors · 2023-11-15
>
> Thanks to the reviewer for the positive review. The reviewer is perfectly right in stating that the main novelty in the paper is to get rid of the linearity assumption and to present results in a very general setup.

---

### Official Review · Reviewer_6832 · 2023-10-31

**Soundness:** 4 excellent
**Presentation:** 4 excellent
**Contribution:** 3 good
**Rating:** 8
**Confidence:** 4

**Summary:**

This paper shows that the dynamics of a residual connection converges that to of an ODE.

First, the authors show that under certain conditions (like the boundedness of the Frobenius norms of the weight matrices) for finite training time as the depth of the network tends to infinity the dynamics of a resnet converge to that of an ODE. This is assuming that the model is trained with clipped gradient descent.

Since here the authors show that the Frobenius norm of the different of the weight matrices is upper bounded by a value that is O(exp(T)), the to show the result for when both T→ \infty and L → \infty, the authors also assume that residual network satisfies a PL condition.

Therefore, the authors show that under these condition as T→ \infty and L→ infty, the dynamics of a residual network tend to an ODE and converges to a function that interpolates the training data.

**Strengths:**

The paper is very well written and easy to follow, where the results and their implications are clearly stated.

The authors show weights of a resnet convert to an ODE while considering the dynamics of the network as well, which is new and interesting.

Given that the result shows that the dynamics of the resnet architecture converges to that of an ODE, and therefore as the authors mention one can utilize existing generalization bounds for neural ODEs etc can under certain conditions now be applied to resnets, given the results that has been shown by the authors.

The authors also show that their results/assumptions hold very well on their synthetic dataset.

For the cifar dataset, the authors show that their results hold when using smooth activations and the dynamics may deviate from neural ODE if one uses a non-smooth activation like ReLU.

**Weaknesses:**

The results are under the condition that the resnet is trained under clipped gradient flow, which is often not used in practice. However, I do understand that the main contribution of the paper is to characterize the behaviour of the model in the limit, so it shouldn’t affect the main contribution of the work.

**Questions:**

For the experiments on cifar, the authors show the plot of a random coefficient and its value as one goes deeper into the layer. However, it would be more beneficial to see the difference in Frobenius norm of the weight matrices as well.

---

> ### Author Response · Authors · 2023-11-15
>
> > The results are under the condition that the resnet is trained under clipped gradient flow, which is often not used in practice. However, I do understand that the main contribution of the paper is to characterize the behaviour of the model in the limit, so it shouldn’t affect the main contribution of the work.
>
> This is a good point—thanks to the reviewer for raising it. Clipping is used in our approach to constraint the gradients to live in a ball. It is merely a technical assumption to avoid blow-up of the weights during training.
>
> However, in any scenario where we know that the weights do not blow up, clipping is not required. A first example of such a scenario is the limit $T \to \infty$ in the paper. In this situation, a Polyak-Lojasiewicz condition holds, and, consequently, the weights remain bounded.
>
> Another scenario is by using gradient flow with momentum (a setup actually closer to Adam, which is very used in practice) instead of vanilla gradient flow. One might then show a similar result to our Theorem 4, without clipping, because the gradient updates in the momentum case are bounded by construction.
>
> We will add a comment about this in the next version of the paper.
>
> > For the experiments on cifar, the authors show the plot of a random coefficient and its value as one goes deeper into the layer. However, it would be more beneficial to see the difference in Frobenius norm of the weight matrices as well.
>
> Thanks to the reviewer for the suggestion, we are currently working on the figure, and will add it to the next version of the paper. We will add a link to the figure in a follow-up comment if time allows before the end of the discussion period.

---

> > ### Author Response · Authors · 2023-11-16
> > **Figure asked by the reviewer**
> >
> > Dear reviewer,
> >
> > We finished the figure according to your suggestions. It can be accessed at the following address:  https://anonymous.4open.science/r/double_blind_implicit_regularization_resnets_nodes-1768/fig_rebuttal.pdf
> >
> > In this figure, we display the average (across layers) of the Frobenius norm of the difference between two successive weights in the convolutional ResNets after training on CIFAR 10, depending on the initialization strategy. The index i corresponds to the index of the convolution layer. Results are averaged over 5 runs. We see that a smooth initialization leads to weights that are in average an order of magnitude smoother than those obtained with an i.i.d. initialization.
> >
> > We will add this figure to the next version of the paper.
> >
> > Best regards,
> >
> > The authors

---

### Official Review · Reviewer_NuSB · 2023-11-09

**Soundness:** 3 good
**Presentation:** 3 good
**Contribution:** 3 good
**Rating:** 6
**Confidence:** 2

**Summary:**

This paper shows that an implicit regularization of deep residual networks towards neural ODEs still holds after training with gradient flow. In particular, this is shown for both large depth (via gradient clipping in finite time) and large time limits (assuming PL condition). Along the way, interesting aspects of the problem are discussed and the results are demonstrated numerically on simple simulations as well as on CIFAR10.

**Strengths:**

The derived results and the neural ODE limits seem rather interesting. The theoretical analyses seem rigorous and their discussions are quite thorough, though I haven't read the proofs in detail. It could be that I have missed a similar related work in the literature, but I do think this seems to be an important contribution, and thus could spark interesting future work on implicit regularization as well as furthering understanding of residual networks via the toolbox of ODEs. Besides, the paper is, in general, very well written, and nicely presented.

**Weaknesses:**

- One of the key drawbacks is requiring weight-tying or weight sharing. While results with weight-tied networks don't look bad, still this would have been nice to take care of. Can the authors elaborate on where all and where exactly does it interfere with the proof strategy?

- The other issue is that the linear overparameterization $m > c_1 n$ would, in practice, be somewhat unreasonable and is thus a bit of a stretch. Maybe this is a norm in the literature, but do the authors have some ideas how to go around this?

**Questions:**

Please see above. Besides:


- Would it be possible to extend the approach to incorporate layer normalization?

---

> ### Author Response · Authors · 2023-11-15
>
> > One of the key drawbacks is requiring weight-tying or weight sharing. While results with weight-tied networks don't look bad, still this would have been nice to take care of. Can the authors elaborate on where all and where exactly does it interfere with the proof strategy?
>
> This is a good point, and the reviewer is right in pointing it out. We consider in the paper a weight-tied initialization in order for the network to satisfy an ODE-like structure at initialization. Such a structure is necessary for an ODE structure to hold after training. As stressed in Section 4.3 of the paper, the weight-tying assumption can be relaxed by taking a so-called “smooth initialization”, where the weights at initialization are a discretization of a smooth (potentially random) function.
>
> It remains that this scheme does not encompass iid initializations (e.g., Gaussian or uniform), which is a classical scheme in initializing neural networks. In this case, the large-depth limit falls into a different regime, namely a stochastic different equation instead of an ordinary differential equation (see Marion et al., 2022, ref. below). It turns out that analyzing the convergence in this regime is challenging and deserves a paper by itself. We will make it clearer in the conclusion of the paper.
>
> Scaling ResNets in the Large-depth Regime, Pierre Marion, Adeline Fermanian, Gérard Biau, Jean-Philippe Vert, arXiv:2206.06929, 2022.
>
> > The other issue is that the linear overparameterization $m > c_1 n$ would, in practice, be somewhat unreasonable and is thus a bit of a stretch. Maybe this is a norm in the literature, but do the authors have some ideas how to go around this?
>
> We thank the reviewer for pointing out this important assumption. The assumption $m > c_1 n$ is required to show that our loss landscape satisfies a Polyak-Lojasiewicz (PL) condition, which is one of the main modern frameworks to show convergence of neural networks. The PL condition can be seen as an alternative to convexity, which is not satisfied in neural networks. Alleviating the assumption $m > c_1 n$ would require working under a weaker condition than PL to show global convergence. Unfortunately, we are not aware of such a weaker condition in the neural network literature at the moment.
>
> Note however, as highlighted in Section 2, that the linear overparameterization condition is mild with respect to usual assumptions in the literature (polynomial overparameterization).
>
> > Would it be possible to extend the approach to incorporate layer normalization?
>
> This is a very interesting suggestion, since normalizations are necessary for neural networks to work in practice. To clarify this point, we propose to add the following discussion to the next version of the paper.
>
> First, conceptually, the $1/L$ normalization introduced in the paper (see, e.g., equation (1)) can be seen as an alternative to normalization (batch or layer). The connections between these three types of normalizations is still theoretically unclear. For instance, batch normalization has been shown empirically to be close to a $1/\sqrt{L}$ normalization (De and Smith, 2020, ref. below).
>
> Second, empirically, the experiment with CIFAR-10 presented in Figure 3 includes batch normalization layers, and the implicit regularization towards a neural ODE still holds in this case.
>
> Finally, mathematically, some of our results extend to a fairly general setting (see Section 4.3) where we only assume that the residual connection is a Lipschitz-continuous $\mathcal{C}^2$ function. The intuition suggests that this should include in particular the case where layer normalizations are added to the architecture, although this should clearly necessitate a rigorous and separate mathematical analysis.
>
> Batch normalization biases residual blocks towards the identity function in deep networks, De and Smith, NeurIPS 2020.

---

### Author Response · Authors · 2023-11-15
**Thanks to the reviewers for the reviews**

Dear reviewers,

We warmly thank you for your time and relevant comments, which will help us improve our work. If accepted, we intend to take into account your suggestions. We answer the specifics of questions pointed out by the reviewers in individual responses.

Sincerely,

The authors

---

### Meta-Review · Area_Chair_q8bF · 2023-12-04

**Metareview:**

This paper demonstrates that initializing ResNets as ODE solutions regularizes such networks toward differentiable solutions. All reviewers agree that the paper is convincing, well-written, and valuable. There is no doubt that it should be accepted.

**Justification For Why Not Higher Score:**

It could also be an oral, actually.

**Justification For Why Not Lower Score:**

The paper is not just correct but also insightful.

---

### Decision · Program_Chairs · 2024-01-16

Accept (spotlight)